# Cardiovirus leader proteins retarget RSK kinases toward alternative substrates to perturb nucleocytoplasmic traffic

**Belén Lizcano-Perret**[1], **Cécile Lardinois**[1], **Fanny Wavreil**[1], **Philippe Hauchamps**[2], **Gaëtan Herinckx**[3], **Frédéric Sorgeloos**[1], **Didier Vertommen**[3], **Laurent Gatto**[2], **Thomas Michiels**[1] *

**1** Molecular Virology unit, de Duve Institute, Université Catholique de Louvain, Brussels, Belgium,
**2** Computational Biology and Bioinformatics unit, de Duve Institute, Université Catholique de Louvain, Brussels, Belgium, **3** MASSPROT platform, de Duve Institute, Université Catholique de Louvain, Brussels, Belgium

* thomas.michiels@uclouvain.be

## Abstract

Proteins from some unrelated pathogens, including small RNA viruses of the family *Picornaviridae*, large DNA viruses such as Kaposi sarcoma-associated herpesvirus and even bacteria of the genus *Yersinia* can recruit cellular p90-ribosomal protein S6 kinases (RSKs) through a common linear motif and maintain the kinases in an active state. On the one hand, pathogens' proteins might hijack RSKs to promote their own phosphorylation (direct target model). On the other hand, some data suggested that pathogens' proteins might dock the hijacked RSKs toward a third interacting partner, thus redirecting the kinase toward a specific substrate. We explored the second hypothesis using the *Cardiovirus* leader protein (L) as a paradigm. The L protein is known to trigger nucleocytoplasmic trafficking perturbation, which correlates with hyperphosphorylation of phenylalanine-glycine (FG)-nucleoporins (FG-NUPs) such as NUP98. Using a biotin ligase fused to either RSK or L, we identified FG-NUPs as primary partners of the L-RSK complex in infected cells. An L protein mutated in the central RSK-interaction motif was readily targeted to the nuclear envelope whereas an L protein mutated in the C-terminal domain still interacted with RSK but failed to interact with the nuclear envelope. Thus, L uses distinct motifs to recruit RSK and to dock the L-RSK complex toward the FG-NUPs. Using an analog-sensitive RSK2 mutant kinase, we show that, in infected cells, L can trigger RSK to use NUP98 and NUP214 as direct substrates. Our data therefore illustrate a novel virulence mechanism where pathogens' proteins hijack and retarget cellular protein kinases toward specific substrates, to promote their replication or to escape immunity.

## Author summary

Nuclear pore complexes (NPCs) are multiprotein complexes forming gates in the nuclear envelope that allow the passage of specific proteins and RNAs between the nuclear and cytoplasmic compartments. Within NPCs, phenylalanine and glycine-rich nucleoporins

**Data Availability Statement:** The mass spectrometry proteomics data have been deposited to the ProteomeXchange Consortium via the

PRIDE partner repository with the dataset identifier PXD034604. MaxQuant outputs are available in the study repository https://github.com/UCLouvain-CBIO/2022-RSK-Nups-VIRO Codes: Processed data and reproducible proDA analyses scripts are available in the study repository at https://github.com/UCLouvain-CBIO/2022-RSK-Nups-VIRO.

**Funding:** BLP and CL were the recipients of FRIA fellowship from the belgian FNRS. Work was supported by the EOS joint programme of Fonds de la recherche scientifique-FNRS and Fonds wetenschappelijk onderzoek-Vlaanderen-FWO (EOS ID: 30981113 and 40007527), Belgian fund for Scientific Research (PDR T.0185.14), Loterie Nationale through support to the de Duve Institute and Actions de Recherches Concertées (ARC) to TM. The funders had no role in study design, data collection and analysis, decision to publish, or preparation of the manuscript.

**Competing interests:** The authors have declared that no competing interests exist.

(FG-NUPs) form a mesh needed to sort specific proteins and RNAs that move in and out of the nucleus. Our previous work showed that some viral and bacterial proteins were able to bind cellular kinases called RSKs and to maintain those kinases in an active state. Here we show that a short protein encoded by small RNA viruses of the genus *Cardiovirus* can act as an adapter molecule, which recruits and directs activated RSK kinases toward the NPCs where they phosphorylate the FG-NUPs. As a consequence, phosphorylated FG-NUPs disrupt the mesh and trigger the uncontrolled diffusion of molecules through the nuclear pore. Perturbation of nucleocytoplasmic trafficking can benefit cardioviruses by preventing the initiation of innate immune responses or by providing nuclear cell components to the cytoplasmic virus replication complexes. Our data support the concept that proteins expressed by pathogens are able to retarget host cell enzymes toward alternative substrates, to the benefit of the pathogen.

## Introduction

Proteins encoded by several unrelated pathogens, including RNA viruses, DNA viruses and bacteria, were recently shown to use a common short linear motif (D/E-D/E-V-F, referred to as DDVF hereafter) to recruit members of the cellular p90-ribosomal S6 protein kinases (RSKs) family: RSK1, RSK2, RSK3 and RSK4 [1,2]. Interestingly, competition and cross-linking experiments, as well as crystallography data showed that these pathogens' proteins, the leader (L) protein (cardioviruses), ORF45 (Kaposi sarcoma-associated herpes virus—KSHV) and YopM (*Yersinia*) use a common interface to recruit RSKs. Binding of the pathogens' proteins prevents RSK dephosphorylation by cellular phosphatases, thereby maintaining RSK in an active state [1,2]. Although infection with all three pathogens leads to RSK activation, the outcome of this activation differs according to the protein bound to RSK. YopM association with RSK was proposed to inhibit the inflammasome and to lead to IL-10 production [3,4]; RSK recruitment by ORF45 enhances lytic replication of KSHV [5,6], whereas RSK recruitment by cardiovirus L protein leads to the inhibition of the antiviral eukaryotic initiation factor 2 alpha kinase 2 (EIF2AK2), better known as PKR [1].

PKR inhibition by L was shown to depend on L interaction with RSK but also on L protein's C-terminal domain. Indeed, the M60V mutation in the C-terminal domain of L does not affect RSK recruitment but abolishes PKR inhibition. These observations led us to propose the "model of the clamp" whereby pathogens' proteins would act as adaptor proteins that force a given enzyme (here RSKs) to act on a specific substrate. Through their DDVF motif, pathogens' proteins recruit and maintain RSKs in an activated state. Through another domain (i.e. the C-terminal domain for L), they recruit proteins that could serve as substrate for RSKs. After phosphorylation, such proteins would act as effectors to the benefit of the pathogen (Fig 1A) [1]. In an alternative model, proteins interacting with RSK through the conserved DDVF motif could directly act as preferential substrates for RSK-mediated phosphorylation (Fig 1B), as was suggested for the cellular protein RHBDF1 (Rhomboid 5 homolog 1) [2].

Cardioviruses belong to the *Picornaviridae* family and include encephalomyocarditis virus (EMCV), Theiler's murine encephalomyelitis virus (TMEV) and the human Saffold virus (SAFV), closely related to TMEV. Despite its very small size (67–76 amino acids) and lack of enzymatic activity, the L protein encoded by these viruses was shown to be multifunctional as, beside inhibiting PKR, it blocks interferon gene transcription, and triggers an extensive diffusion of nuclear and cytoplasmic proteins across the nuclear membrane [1,7–16]. This nucleo-cytoplasmic trafficking perturbation was associated with the hyperphosphorylation of

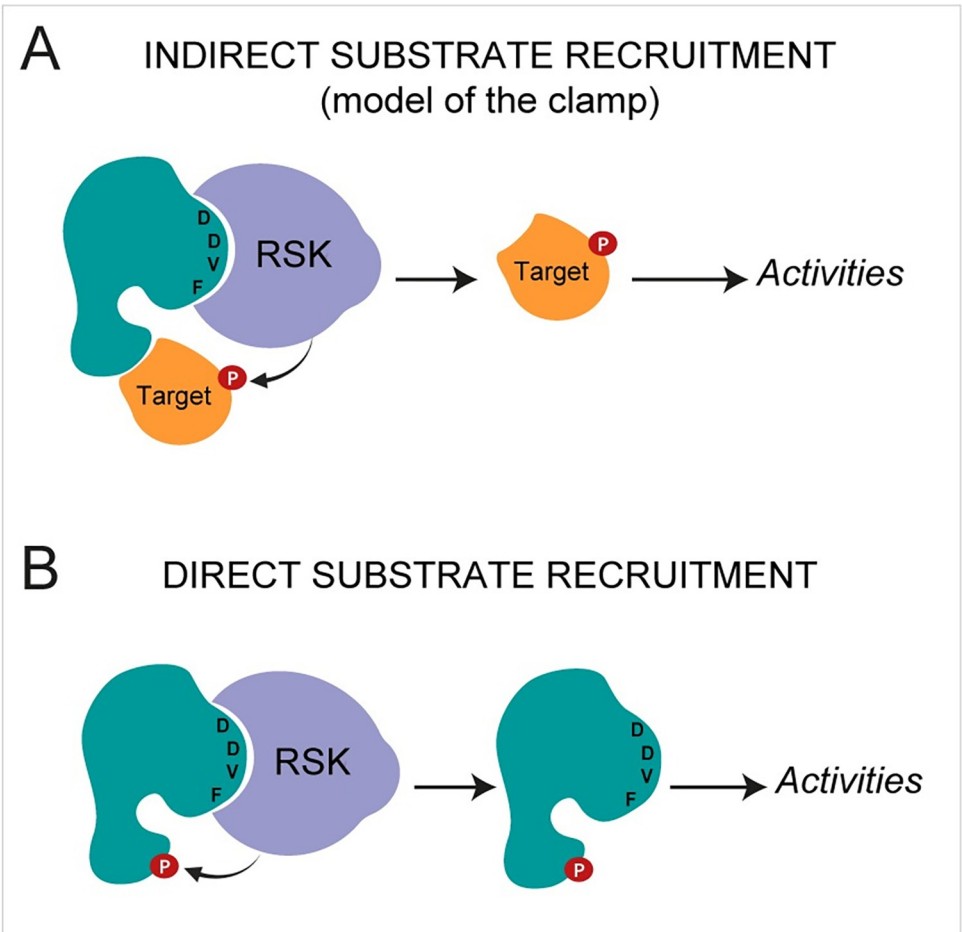

**Fig 1. Pathogen's proteins hijack RSK kinases.** (A) Indirect substrate recruitment—model of the clamp: a pathogen's protein acts as an adaptor protein, which binds and activates RSK through its DDVF motif and recruits a target protein to be phosphorylated by RSK through another domain. (B) Direct substrate recruitment. A protein interacts with RSK through its DDVF motif and is directly phosphorylated by RSK.

phenylalanine-glycine-nucleoporins (FG-NUPs) such as NUP98, NUP153 or NUP62 [17–20]. FG-NUPs are proteins of the nuclear pore complex (NPC) that possess intrinsically disordered domains, rich in phenylalanine and glycine residues. These domains form a mesh in the center of the nuclear pore, which enables their interaction with karyopherins, thus allowing selective transport of proteins and RNA through the NPC [21–24]. Molecular dynamic simulations predict that FG-NUP phosphorylation drastically decreases the density of FG-NUPs inside the pore [25]. Accordingly, electron microscopy analysis of EMCV-infected cells displayed such a density loss in the center of the NPC [20]. Through perturbation of the NPC, cardioviruses trigger a diffusion of nuclear proteins to the cytoplasm which can be used by the virus for viral replication and translation [16].

Preliminary data suggested that nucleocytoplasmic trafficking perturbation by cardiovirus L proteins depended on RSK and on the recruitment of a cellular target by the L-RSK complex (model of the clamp). The aim of this work was to assess whether RSK could indeed be retargeted by L to new substrates in infected cells and to identify such a substrate that could trigger nucleocytoplasmic trafficking perturbation. Our data show that L can recruit both RSK and FG-NUPs and that FG-NUPs can act as direct RSK substrates in infected cells. This work provides strong support to the model of the clamp and elucidates a novel virulence mechanism.

## Results

### Cardiovirus L-mediated nucleocytoplasmic trafficking perturbation depends on RSK recruitment by L

We tested whether L-mediated redistribution of nuclear and cytoplasmic proteins across the nuclear envelope and the hyperphosphorylation of FG-NUPs depends on both RSK and L. In agreement with previous reports [14], cytosolic diffusion of the nuclear protein PTB occurred in HeLa cells infected for 10 hours with TMEV expressing a wild type L protein ($L^{WT}$) but not in non-infected cells or in cells infected with viruses carrying the M60V mutation in the C-terminal part of L ($L^{M60V}$)(Fig 2A and 2B). In the latter cells, PTB however partly appeared as cytosolic punctae, likely corresponding to stress granules that were shown to form in the absence of PKR inhibition by L [12]. Importantly, $L^{WT}$-mediated PTB diffusion was almost abrogated in RSK-deficient HeLa cells (HeLa-RSK-TKO) (Fig 2B and 2C). Viral replication appeared to be decreased in the absence of RSK, in agreement with increased PKR activation in these cells (Fig 2D). Interestingly, PTB diffusion was restored after transduction of HeLa-RSK-TKO cells with lentiviruses expressing any of the four human RSK isoforms (Fig 2B–2C). Accordingly, NUP98 hyperphosphorylation was dramatically decreased in HeLa-RSK-TKO cells and restored in cells transduced to express any of the four RSK isoforms (Fig 2D). Similar observations were made in the case of EMCV as soon as 5 hours post-infection, although dependence on RSK was less pronounced (Fig 2E–2H). Thus, PTB diffusion out of the nucleus and NUP98 hyperphosphorylation depend on both L and RSK.

In addition, TMEV and EMCV expressing $L^{WT}$ but not mutant L proteins triggered an RSK-dependent disruption of nucleocytoplasmic trafficking of other proteins harboring canonical nuclear export (NES) and import (NLS) sequences, as shown by diffusion of GFP-NES and RFP-NLS proteins in live cells that stably express these proteins (Figs 3A–3D and 4). This observation demonstrates that nucleocytoplasmic redistribution of proteins was not specific to PTB. We noticed that diffusion of RFP-NLS was less dependent on RSK than diffusion of GFP-NES (Fig 3A and 3C). As for PTB diffusion, RSK dependence of protein redistribution was less prominent for EMCV than for TMEV.

Ectopic expression of TMEV $L^{WT}$ but not the $L^{F48A}$ mutant (affected in the RSK-binding motif) was sufficient to trigger nucleocytoplasmic redistribution of proteins (Fig 3E–3G). Also, expression of $L^{M60V}$ or YopM did not affect nucleocytoplasmic trafficking although these proteins can activate RSK [1]. Taken together, these data show that L-mediated nucleocytoplasmic trafficking alteration does not dependent on virus replication per se or on the expression of other viral proteins and requires both interaction with RSK and, at least for TMEV L, integrity of the C-terminal part of the protein.

### Identifying RSK partners in TMEV-infected cells using BioID2

According to the model of the clamp, TMEV L would recruit RSK through its central DDVF motif and would recruit potential RSK substrates through its C-terminal domain. As suggested above, the M60V mutation of L would prevent the recruitment of such a target, without affecting RSK binding.

In order to identify target proteins that are recruited by L as potential RSK targets, we used the modified bacterial biotin ligase BioID2 (BioID) which, when fused to a protein of interest, allows promiscuous biotinylation of proximal proteins in the cell [26].

A BioID-RSK2 (BioID-RSK) fusion was constructed and expressed in RSK1/2-deficient HeLa cells (HeLa-RSK-DKO) by lentiviral transduction to identify RSK's potential partners during TMEV infection (Fig 5A). After testing that the BioID-RSK construct was activated

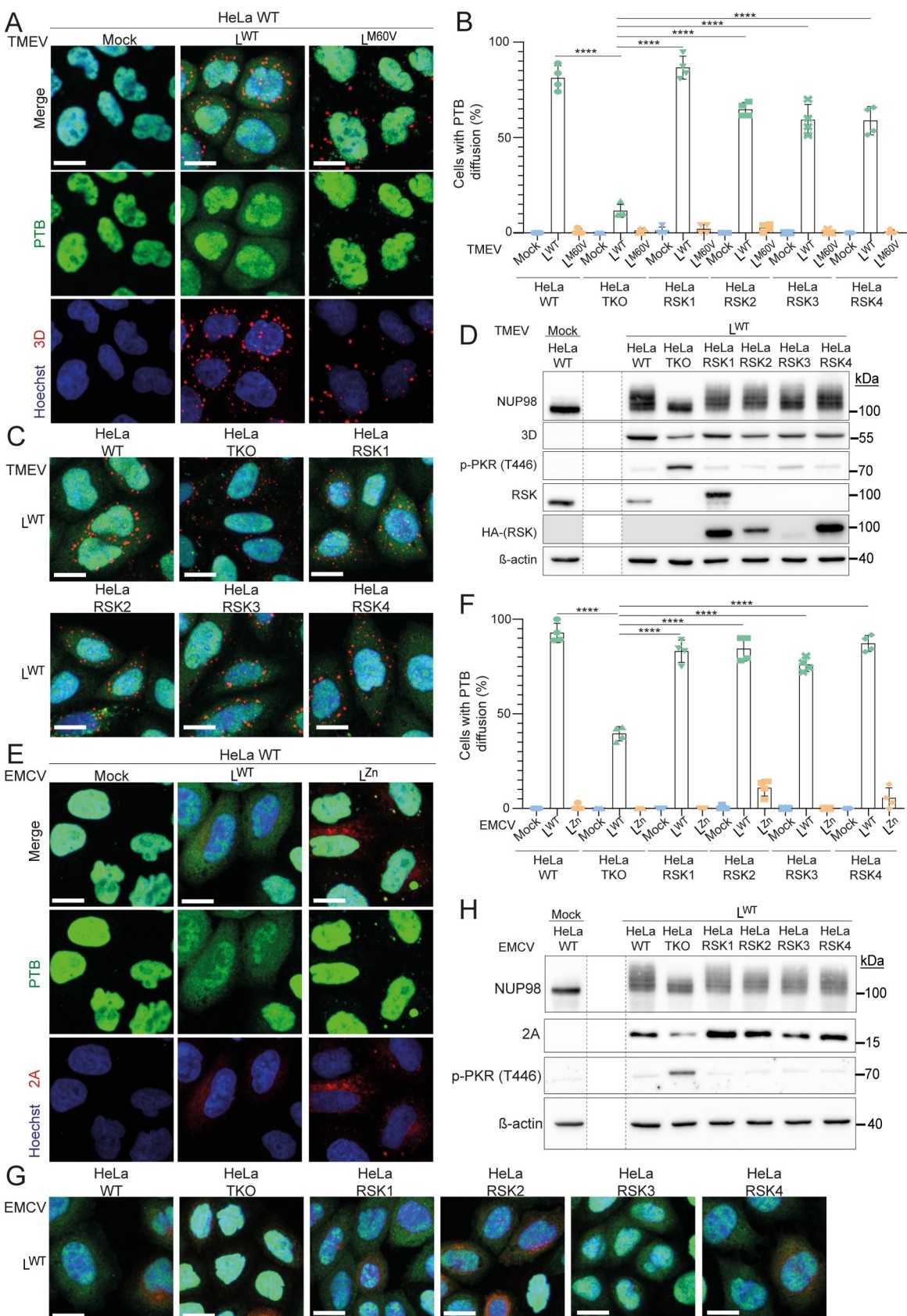

**Fig 2. PTB diffusion and NUP98 hyperphosphorylation induced by cardioviruses depends on both the L protein and RSK kinases.** (A-C and E-G) PTB diffusion out of the nucleus depends on both L and RSK. (A and E) Confocal microscopy images of HeLa WT cells infected with TMEV for 10h (A) or with EMCV for 5h (E). (B and F) Graphs showing the percentage of cells with PTB diffusion (mean ± SD), among cells infected by TMEV (3D-positive) for 10h (B) or by EMCV (capsid-positive) for 5h (F). Counts: 25–50 infected cells per experiment (n = 4). One-way ANOVA was used to compare all samples with each other. Shown are significant differences observed for $L^{WT}$-infected samples between HeLa-RSK-TKO cells transduced with an empty vector (HeLa TKO) and cells re-expressing indicated RSK isoforms. (C and G) Confocal microscopy images of HeLa-WT or HeLa RSK-TKO cells transduced with an empty vector (HeLa TKO) or re-expressing the indicated RSK isoforms. Cells were infected with TMEV for 10h (C) or EMCV for 5h (G). 3D and 2A were stained as controls of TMEV and EMCV infection respectively. Scale bar: 20μm (D and H) Western blots showing the RSK-dependent hyperphosphorylation of NUP98 (shift upwards) or the inhibition of PKR activation induced by TMEV (D) or EMCV (H). 3D or 2A were detected as control of infection, and ß-actin as loading control. RSK re-expression in RSK-TKO cells was monitored with anti-HA (detection of HA-RSK1, -2, -3, -4) and anti RSK (detecting RSK1) antibodies. Dashed lines between lanes indicate deletion of an irrelevant lane from the same membranes.

(i.e. phosphorylated) in the presence of L in infected cells and able to rescue L activities such as L-mediated PKR inhibition (S1A Fig), we performed biotinylation experiments in order to identify RSK partners in the context of infected cells. HeLa BioID-RSK cells were infected with either $L^{WT}$ or $L^{M60V}$ viruses for 16 hours in the presence of biotin. Biotinylated proteins were pulled down using streptavidin beads and processed for mass spectrometry analysis. Western blot detection of biotinylated proteins (Fig 5B) confirmed the concentration of biotinylated proteins in the pulled down fraction and revealed the presence of a 100kDa band specific to the $L^{WT}$-infected sample, suggesting that the L protein indeed influenced target biotinylation by the BioID-RSK fusion.

## Identifying L protein partners using BioID

Several attempts to identify proteins that are recruited by TMEV L were made in our lab by coimmunoprecipitation. However, beside RSK, no clear L partner was consistently identified. It is not unlikely that proteins recruited by L would interact with L only transiently and with low affinity. We thus used the BioID system to identify such proteins that may transiently interact with L in infected cells and then serve as RSK substrates. However, due to packaging limitations in picornaviruses, the BioID coding sequence (696 nt) would be too long to be accommodated into the viral genome. We thus generated *trans*-encapsidated replicons where a sequence coding for BioID-L and eGFP was substituted for the capsid-coding region (Fig 5C). These replicons were encapsidated in 293T cells stably expressing a capsid coding sequence carrying synonymous mutations in the VP2 region that affect *CRE* function [27] to prevent the selection of replication-competent wild type viruses that might emerge through recombination. *Trans*-encapsidated replicons coding for BioID fused to $L^{WT}$, $L^{M60V}$ and $L^{F48A}$ were produced (Fig 5A). In HeLa cells infected with these replicons, L activities (i.e. RSK phosphorylation, PKR inhibition and NUP98 hyperphosphorylation) were as expected (S1B Fig). HeLa cells were then infected with BioID-L replicons for 14 hours in the presence of biotin. Biotinylated proteins were pulled down in stringent conditions. Western blot confirmed protein biotinylation in infected cell extracts and showed that some bands occurred in $L^{WT}$ and $L^{F48A}$ but not in $L^{M60V}$ samples (Fig 5D). These results confirm specific biotinylation of proteins by BioID-L.

## FG-NUPs are enriched in BioID-L and BioID-RSK proxeomes

We expected that RSK targets recruited by L would be present in both BioID-RSK and BioID-L screens. Pulled down biotinylated proteins from both screens (n = 3 for each screen) were identified by mass spectrometry (MS) and sorted according to their peptide spectrum match (PSM) number (for more detail on the calculations see material and methods). For

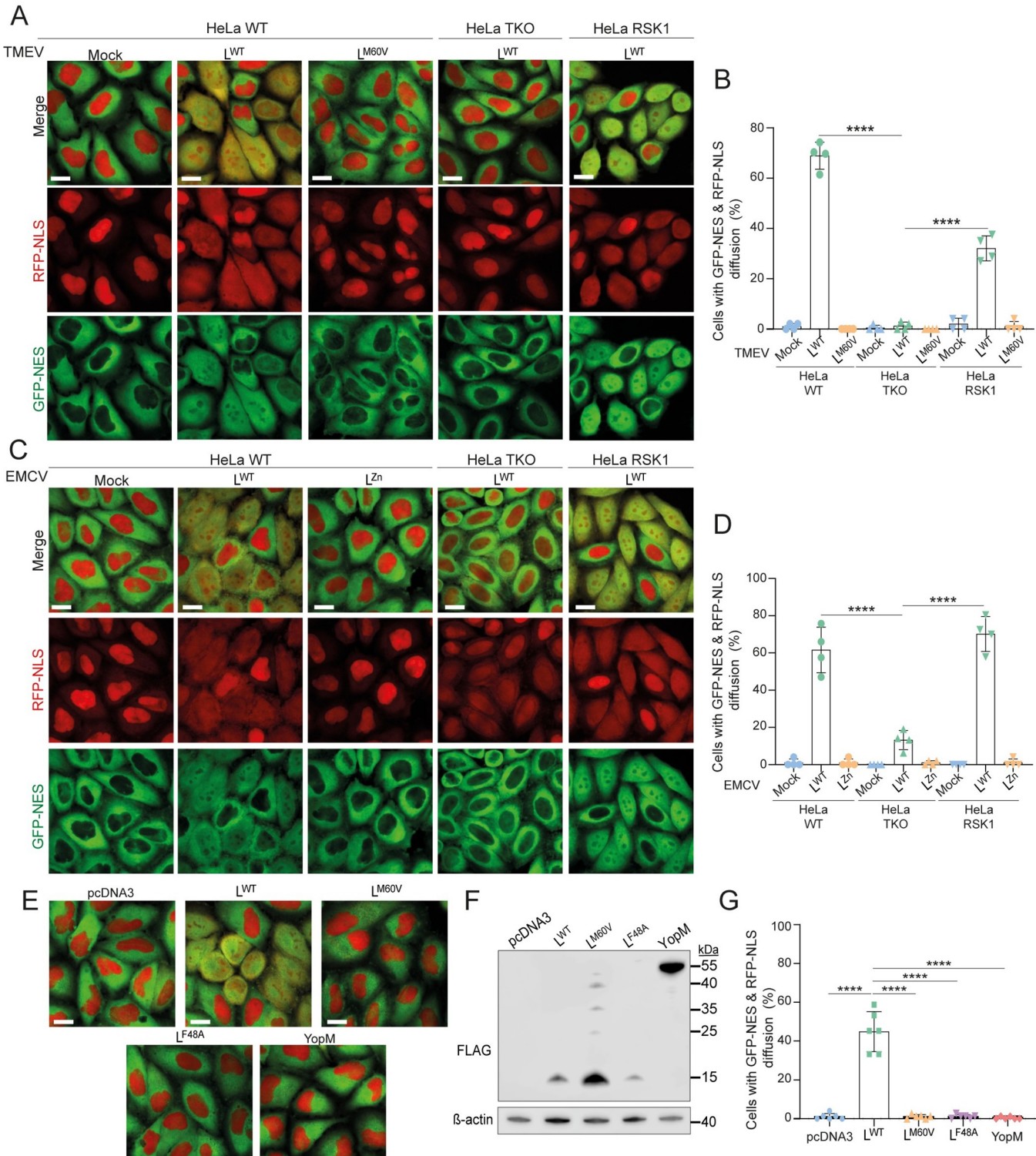

**Fig 3. Nucleocytoplasmic traffic perturbation in live cells depends on both the L protein and RSK kinases.** (A-D) Graphs and confocal microscopy images showing the quantification of RFP-NLS and GFP-NES diffusion (mean ± SD) in live HeLa-LVX cells infected with TMEV for 10h to 11h (A-B) or EMCV for 4h30 to 5h30 (C-D). HeLa-LVX cells were either WT, or RSK-TKO cells transduced with an empty vector (HeLa TKO) or re-expressing RSK1 (HeLa RSK1). Focus was set on the red channel and diffusion of NES-GFP was evaluated on merge images. 50 cells per experiment (n = 4) were counted and considered diffusion-positive when green fluorescence was as prominent as red fluorescence in the nucleus. (E-G) Impact of L on nucleocytoplasmic traffic. HeLa-LVX cells transfected with an empty plasmid or with plasmids expressing L^WT, L mutants or YopM. (E) Confocal microscopy of HeLa-LVX cells 24hours post-

transfection. (F) Western blot of HeLa-LVX cells 24h post-transfection. FLAG detection shows the expression of (FLAG-)L proteins and (FLAG-)YopM. (G) Graph showing the percentage of cells with GFP-NES and RFP-NLS diffusion (mean ± SD). Counts: 70±5 cells per experiment (n = 6). One-way ANOVA was used to compare all samples with each other. Shown are significant differences observed between all samples and $L^{WT}$. Scale bar: 20μm.

BioID-RSK, proteins were sorted according to their abundance in $L^{WT}$-infected cells relative to that in non-infected cells and in cells infected with the $L^{M60V}$ mutant virus (Fig 6A, vertical axis). For BioID-L (replicons), proteins were sorted according to their abundance in $L^{WT}$ and $L^{F48A}$ samples relative to that in $L^{M60V}$-infected and in non-infected samples (Fig 6A, horizontal axis). Hence, the proteins that would be recruited by the C-terminal domain of L are those that have high ratios in both screens. Interestingly, many of these proteins were FG-NUPs (Fig 6A and 6B). ProDA (Probabilistic Dropout Analysis) statistical analysis [28] was used to identify proteins whose abundance differs in pairwise comparisons: $L^{WT}$ versus $L^{M60V}$ for the BioID-RSK experiments, and $L^{WT}$ versus $L^{M60V}$ or $L^{F48A}$ versus $L^{M60V}$ for the BioID-L experiment. The table in Fig 6B shows adjusted P-values obtained from this analysis for the 20 best-ranked proteins. In the BioID-RSK experiments, FG-NUPs had the lowest adjusted P-values (though not significant) and highest fold change (Fig 6B and 6C). In the BioID-L experiments many FG-NUPs reached significant scores when pairwise comparisons were made between $L^{WT}$ and $L^{M60V}$ or between $L^{F48A}$ and $L^{M60V}$ (Fig 6B and 6C). Most importantly, FG-NUPs that had previously been shown to be hyperphosphorylated during cardiovirus infection: NUP62, NUP98, NUP153 and NUP214 [17–20] exhibited statistical significance (Fig 6B). Lower significance exhibited by BioID-RSK compared with the BioID-L data likely reflects the fact that only part of the BioID-RSK molecules are retargeted by L toward new partners. From these results we conclude that L recruits RSK through its DDVF motif and then docks RSK toward FG-NUPs when the C-terminal domain of L is intact.

## Proteins biotinylated by BioID-RSK and BioID-L are localized at the NPC

We analyzed the subcellular localization of proteins that were biotinylated by BioID-RSK and by BioID-L in infected cells. To this end, biotinylation experiments were performed as above but infected cells were fixed after 10h of infection, permeabilized and stained them with Alexa Fluor-conjugated streptavidin. For BioID-RSK, mock-infected cells showed staining of biotinylated proteins in the nucleus, in agreement with the mostly nuclear localization of RSKs [29] (Fig 7A–7C). In cells infected with the $L^{WT}$ virus, biotinylated proteins formed a faint rim around the nucleus that colocalizes with the FG-NUPs POM121 and NUP98, in addition to the diffuse nuclear staining, showing that part of the (BioID)-RSK molecules are recruited to the nuclear envelope. In cells infected with the $L^{M60V}$ or $L^{F48A}$ mutant viruses, this nuclear rim was lost, in agreement with a lack of $L^{M60V}$ interaction with nucleoporins and the lack of $L^{F48A}$ interaction with BioID-RSK.

For BioID-L replicons, staining occured in the nucleus and at the nuclear envelope for $L^{WT}$, in agreement with an interaction of L with both RSK and NUPs (Fig 8A–8C). Proteins biotinylated by BioID-$L^{M60V}$ showed diffuse staining in the nucleus, in agreement with $L^{M60V}$ interaction with RSK but not with nucleoporins. Proteins biotinylated by BioID-$L^{F48A}$ showed extensive colocalization with nucleoporin POM121 at the nuclear envelope (Fig 8C), in agreement with the lack of interaction with RSK, which would likely sequester part of the L protein in the nucleoplasm. In conclusion, labeling of biotinylated proteins supports our model where L interacts with RSK in the nucleus and recruits a portion of RSK molecules to the NPC, via an interaction between the C-terminal domain of L and NUPs.

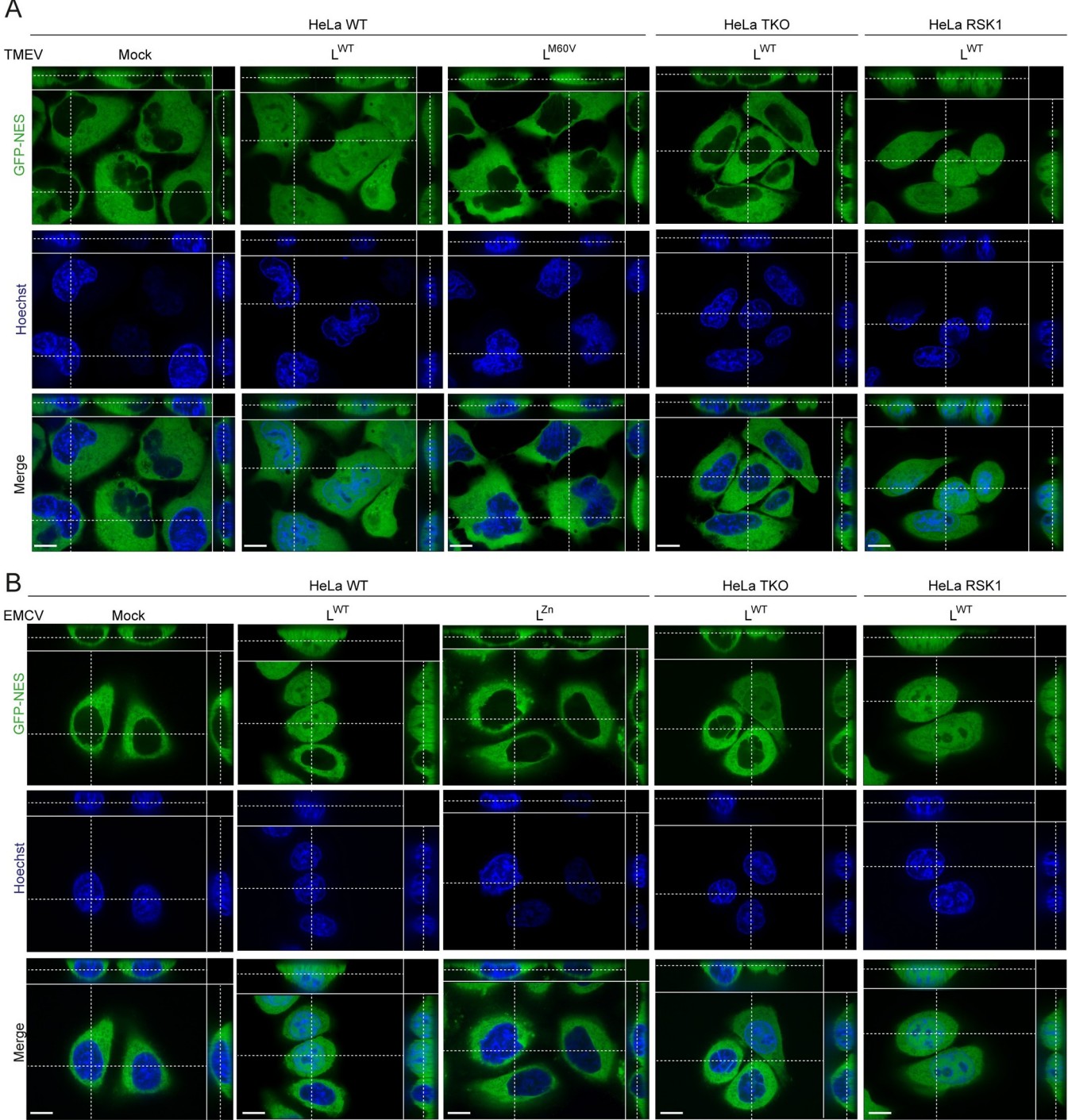

**Fig 4. GFP-NES diffusion in TMEV and EMCV infected cells.** (A-B) Z-stack confocal microscopy images showing GFP-NES diffusion in live HeLa-LVX cells infected with TMEV for 10h to 11h (A) or EMCV for 4h30 to 5h30 (B). HeLa LVX cells were either WT, or RSK-TKO cells transduced with an empty vector (HeLa TKO) or re-expressing RSK1 (HeLa RSK1). Scale bar: 10μm.

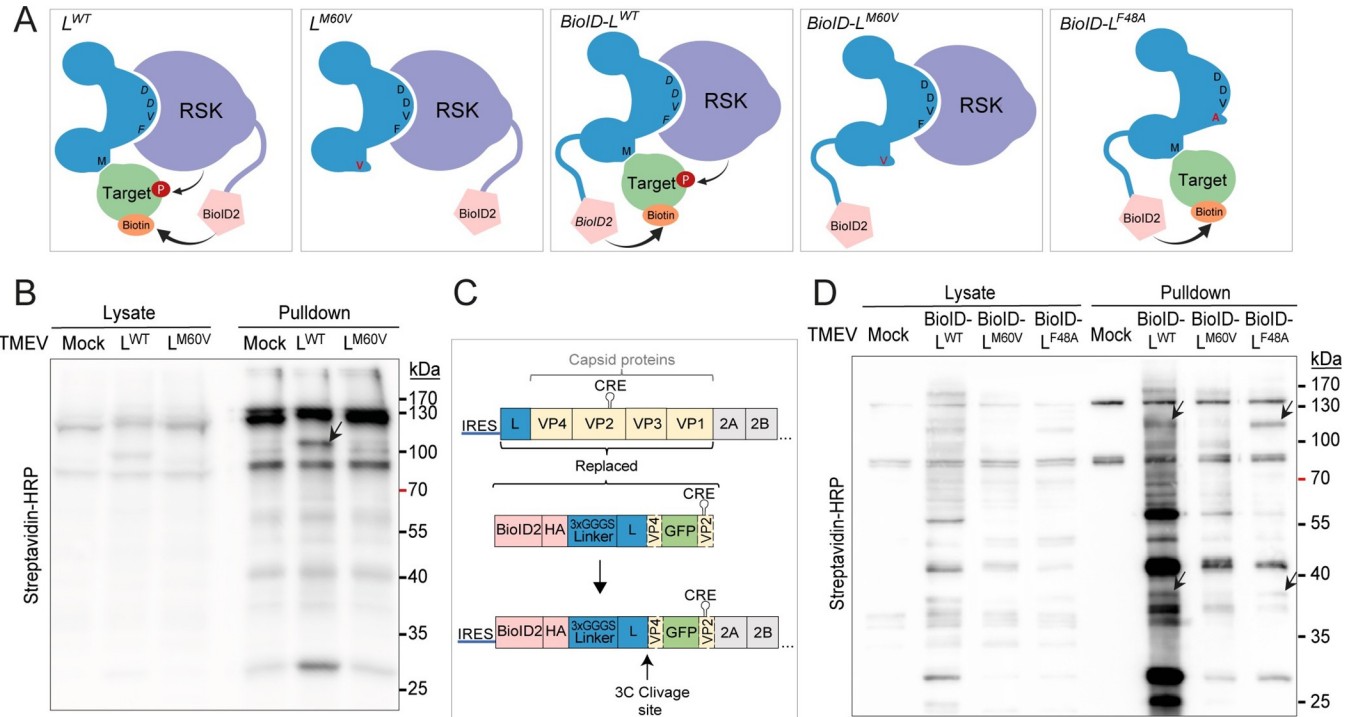

**Fig 5. BioID-RSK and BioID-L fusion proteins biotinylate specific proteins during cardiovirus infection.** (A) Cartoon showing the expected biotinylation by BioID-RSK or BioID-L fusion proteins of a target recruited by the C-terminal domain of L. (B) Western blot of proteins biotinylated by BioID-RSK in lysates and pulled down samples of TMEV-infected cells. HeLa BioID-RSK cells were incubated for 2 days without biotin. Cells were then infected for 16h (MOI 2.5) with $L^{WT}$ or $L^{M60V}$ viruses in medium containing 5μM biotin. Biotinylated proteins were pulled-down using streptavidin-magnetic beads. (C) Schematic representation of BioID-L TMEV replicon constructs. Capsid protein sequences were replaced by BioID-L and GFP sequences. The beginning of the VP4 sequence was kept in order to allow BioID-L protein processing by viral protease 3C, and the CRE sequence (localized in VP2) was re-introduced after GFP in the construct to allow replication. (D) Western blot of proteins biotinylated by BioID-L in lysates and pulled down samples of TMEV infected cells. HeLa cells were infected for 14h (MOI 2.5) with BioID-$L^{WT}$, BioID-$L^{M60V}$ or BioID-$L^{F48A}$ replicons in medium containing 5μM biotin. Biotinylated proteins were then pulled-down using streptavidin-magnetic beads.

## FG-NUPs are direct substrates for RSK in cardiovirus-infected cells

Since FG-NUPs are known to be hyperphosphorylated during cardiovirus infection and since our results showed that RSK is in their close proximity, we aimed to test if RSK can directly phosphorylate these FG-NUPs in infected cells. For this, we used the « analog-sensitive kinase system » developed by the group of Shokat [30]. This system is based on the use of an ATP analog: N6-alkylated ATP-γ-S (A*TP-S), bulkier than ATP, that only fits in ATP binding pockets of kinases that have been mutated to accommodate such bulkier analogs. The advantage of this technique is that the mutated kinase (RSK in our case) is the only kinase in the cell that can use A*TP-S and thereby thiophosphorylate its substrates. After alkylation, such substrates can be recognized using a specific anti-thiophosphate ester antibody (Fig 9A). We identified Leu147 as the RSK2 gatekeeper residue (i.e. residue in the ATP-binding pocket that, when mutated to a smaller residue, allows the access of the A*TP-S) by homology to Thr338 in the consensus sequence defined for c-SRC [31] (Fig 9B). Leu147 was mutated either to Gly (As1-RSK) or to Ala (As2-RSK). To test for ATP analog usage by the modified RSK kinases, WT, As1- and As2-RSKs were stably expressed in HeLa-RSK-TKO cells by lentiviral transduction, yielding WT-, As1- and As2-RSK cells. Expression of both As1- and As2-RSK allowed L functions (i.e. PKR inhibition, RSK activation and NUP98 hyperphosphorylation) (S2A Fig). The ATP analog N6-Bn-ATP-γ-S was better incorporated than N6-PhEt-ATP-γ-S by the As-RSKs (S2B

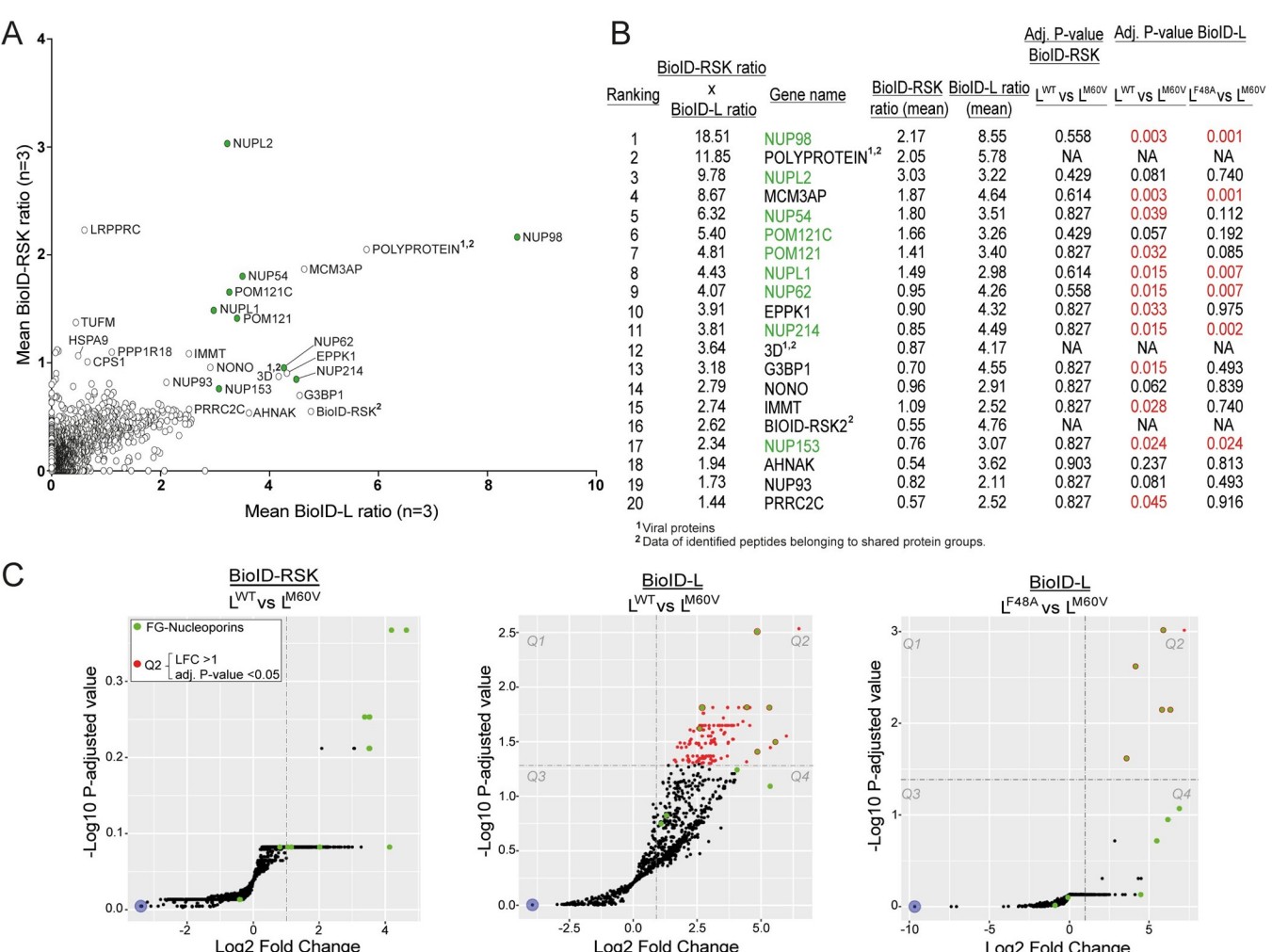

**Fig 6. BioID-RSK and BioID-L proxeomes identify FG-NUPs as targets recruited by the L-RSK complex.** (A) Proteins identified in the proximity of BioID-RSK (Y axis) and BioID-L (X axis). Y axis shows proteins detected after infection with $L^{WT}$ but not $L^{M60V}$ viruses and in non-infected conditions (= PSMs in $L^{WT}/(L^{M60V}+NI+1)$). X axis shows proteins detected after infection with BioID-$L^{WT}$ and BioID-$L^{F48A}$ but not BioID-$L^{M60V}$ replicons and in non-infected conditions (= PSMs in $(L^{WT}+L^{F48A})/(L^{M60V}+NI+1)$). FG-NUPs are identified by green dots. (B) Table showing the adjusted P-values for the 20 best ranked proteins. Ranking was attributed by multiplying the BioID-RSK ratio by the BioID-L ratio. Statistical analysis of pairwise comparisons made between $L^{WT}$ and $L^{M60V}$ for BioID-RSK and between $L^{WT}$ and $L^{M60V}$ or $L^{F48A}$ and $L^{M60V}$ for BioID-L. Adjusted P-values < 0.05 are colored in red, FG-NUPs are colored in green. (C) Volcano plots showing the same pairwise comparisons as in B. Proteins having an adjusted P-value < 0.05 and a $Log_2$ fold change (LFC) > 1 are colored in red (= proteins in Q2). FG-NUPs are colored in green.

Fig). The As2-RSK kinase was better expressed than As1-RSK and readily accommodated this analog (S2A and S2B Fig). Therefore, As2-RSK cells and N6-Bn-ATP-γ-S were chosen for further experiments. HeLa WT-RSK or As2-RSK cells were then infected with $L^{WT}$ and $L^{M60V}$ viruses for 8h30min. Cells were then permeabilized with digitonin and treated with A*TP-S (N6-Bn-ATP-γ-S) for 1 hour. Then, NUP98 was immunoprecipitated and its thiophosphorylation status was analyzed, after alkylation, by western blot with the anti-thiophosphate ester antibody. The blot shown in Fig 9C shows a band highly enriched in the immunoprecipitated sample, migrating at the expected molecular mass for NUP98 (98kDa). This band was most prominent in the samples from As2-RSK cells, confirming the specificity of thiophosphate ester detection. Importantly, this band was detected much more in samples of cells infected with the $L^{WT}$ virus than in samples of cells that were not infected or infected with the $L^{M60V}$

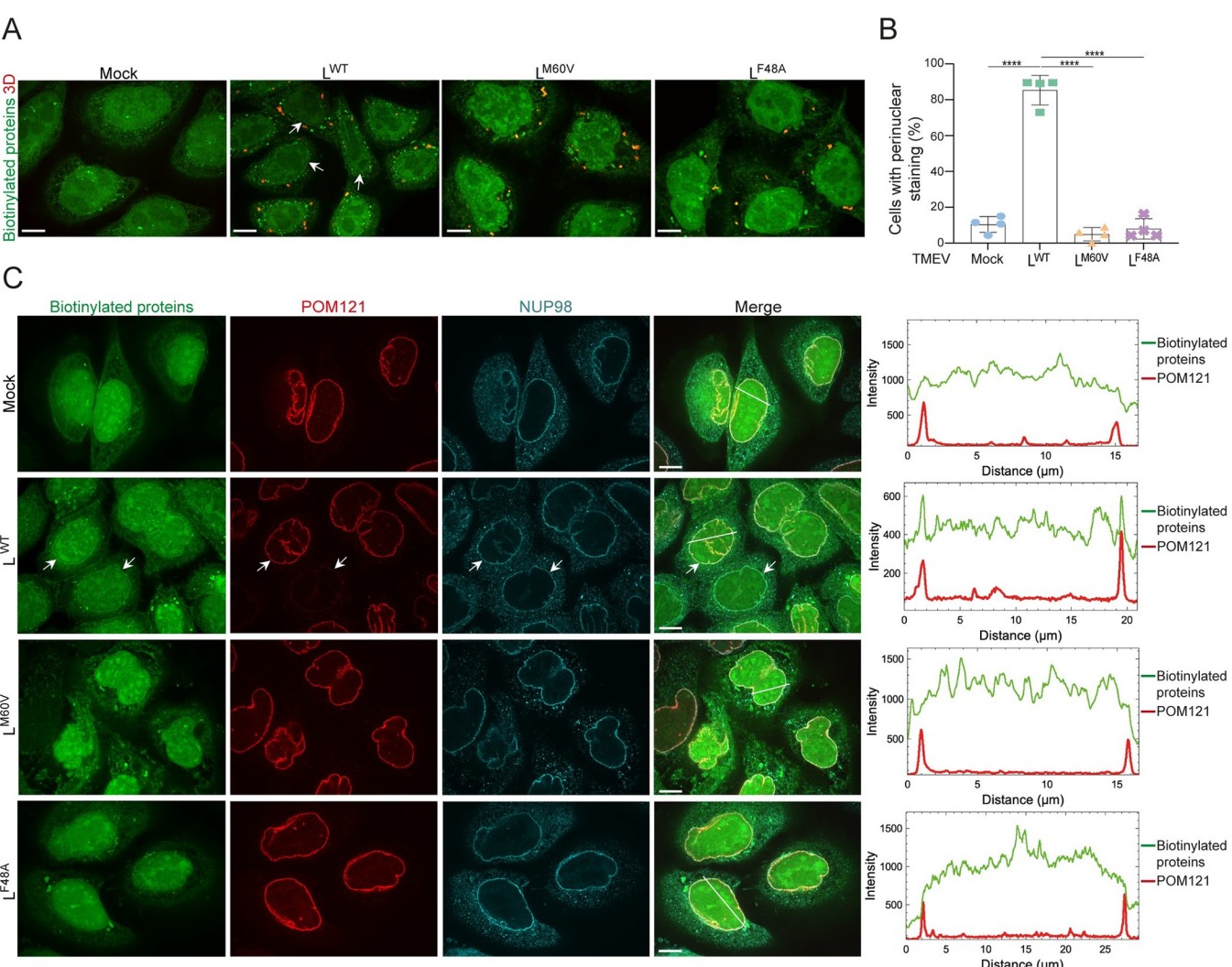

**Fig 7. Subcellular localization of proteins biotinylated by BioID-RSK.** (A) Confocal microscopy images of biotinylated proteins (green) and 3D viral polymerase (red) detected in BioID-RSK expressing HeLa cells infected with $L^{WT}$, $L^{M60V}$ or $L^{F48A}$ viruses for 10h (MOI 5). 3D viral polymerase was stained as a control of infection. (B) Quantification of cells showing a visible nuclear rim of biotinylated proteins in HeLa BioID-RSK cells infected with $L^{WT}$, $L^{M60V}$ or $L^{F48A}$ viruses (mean ± SD). Counts: 50 ± 10 infected cells per experiment (n = 4). One-way ANOVA was used to compare all samples with each other. Shown are significant differences with $L^{WT}$. (C) BioID-RSK biotinylates proteins that colocalize with the FG-NUPs POM121 and NUP98. Confocal microscopy images and intensity vs distance plots of biotinylated proteins (green), POM121 (red), and NUP98 (cyan) in HeLa BioID-RSK cells infected with $L^{WT}$, $L^{M60V}$ or $L^{F48A}$ viruses in parallel to cells shown in panel A. White arrows point to examples of cells exhibiting a nuclear rim staining. Scale bar: 10µm.

virus. These data indicate that NUP98 can be directly phosphorylated by RSK and that this phosphorylation only occurs after infection with the $L^{WT}$ virus. For EMCV, as this virus replicates much faster than TMEV, NUP98 was immunoprecipitated after 3h30 of infection. Thiophosphorylation of immunoprecipitated proteins was tested by western blot using the anti-thiophosphate ester antibody (Fig 9D). As for TMEV, a ~100kDa protein likely corresponding to NUP98 was thiophosphorylated by As2-RSK only in $L^{WT}$ infected conditions. It is noteworthy that additional bands were detected, which likely correspond to other FG-NUPs such as NUP62, NUP153 and NUP214 that coimmunoprecipitated with NUP98.

In order to confirm FG-NUPs thiophosphorylation by As2-RSK in infected cells, experiments were performed the other way around. Therefore, thiophosphorylated proteins were immunoprecipitated from cells infected with TMEV variants, using the thiophosphate ester

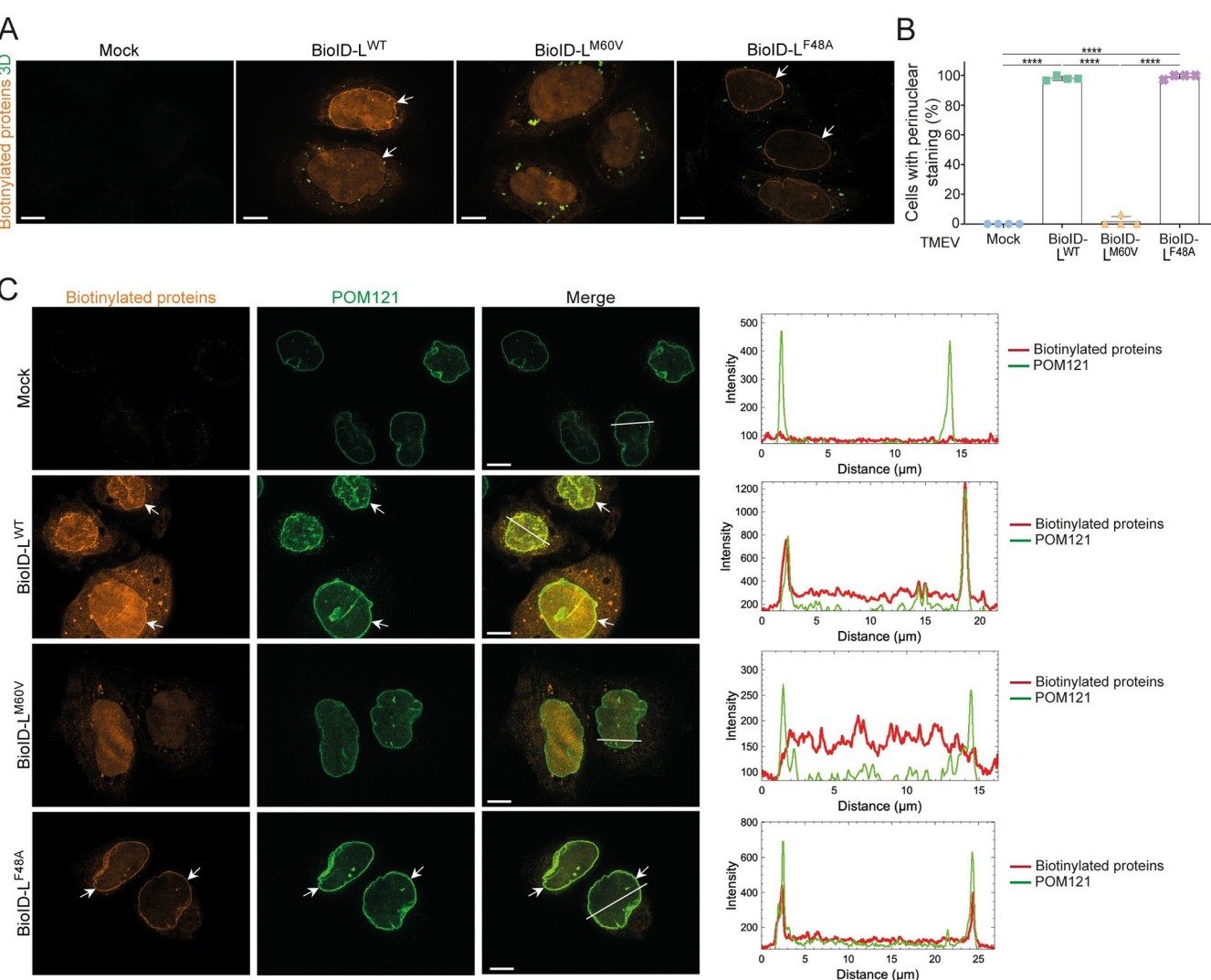

**Fig 8. Subcellular localization of proteins biotinylated by BioID-L.** (A) Confocal microscopy images of biotinylated proteins stained with streptavidin-Alexa Fluor 594 (orange) and 3D viral polymerase (green) in HeLa cells infected for 10h with BioID-L$^{WT}$, BioID-L$^{M60V}$ or BioID-L$^{F48A}$ replicons (MOI 5). 3D viral polymerase was stained as a control of infection. (B) Quantification of nuclear rim staining in HeLa cells infected with BioID-L$^{WT}$, BioID-L$^{M60V}$ or BioID-L$^{F48A}$ viruses (mean ± SD). Counts: 31 ± 11 infected cells per experiment (n = 4). One-way ANOVA was used to compare all samples with each other. Significant differences are shown. (C) BioID-L biotinylates proteins that colocalize with the FG-NUP POM121. Confocal microscopy images and intensity vs distance plots of biotinylated proteins (orange) and POM121 (green) in HeLa cells infected with BioID-L$^{WT}$, BioID-L$^{M60V}$ or BioID-L$^{F48A}$ viruses for 10h (MOI 5). White arrows point to examples of cells exhibiting a nuclear rim staining. Scale bar: 10μm.

antibody. Then the IP fraction (i.e. thiophosphorylated proteins) was analyzed by western blot for the presence of NUP98 (Fig 9E) and NUP214 (S2C Fig). Again, these NUPs were most prominently detected in the conditions where the L protein was WT and RSK was mutated to accomodate the ATP analog (As2-RSK). These data confirm that RSK directly phosphorylates FG-NUPs when the L$^{WT}$ protein is present.

Taken together, these results confirm that part of the cellular RSKs can be redirected to the nuclear pore complex by the L protein, where RSKs phosphorylate FG-NUPs. By interacting with RSK through its DDVF motif and with FG-NUPs through its C-terminal domain, L serves as a viral adapter protein to modulate RSK's activity and redirects these kinases toward new substrates thus supporting the model of the clamp. Hyperphosphorylation of FG-NUPs

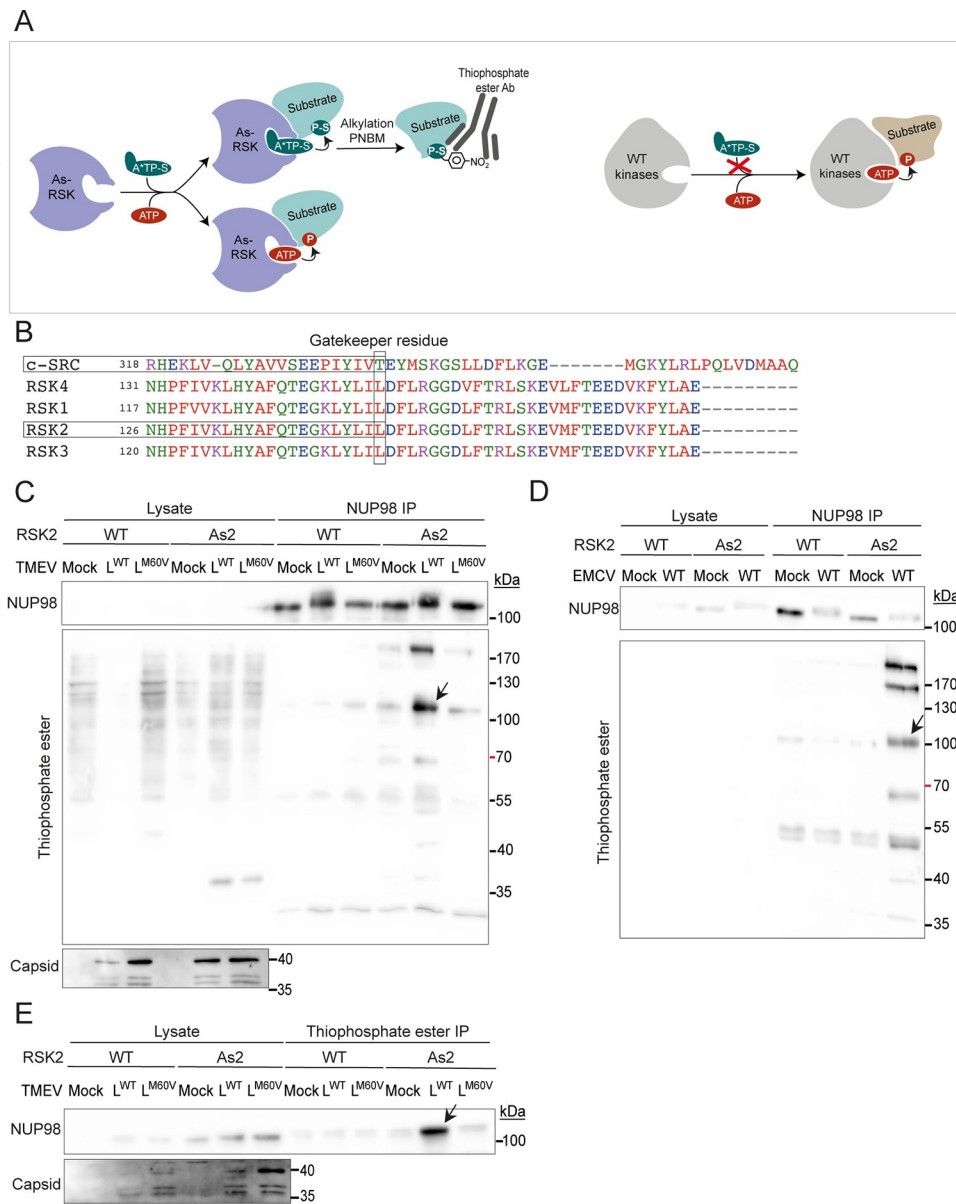

**Fig 9. RSK directly phosphorylates NUP98 in TMEV and EMCV infected cells.** (A) Cartoon showing the principle of the analog-sensitive kinase system. The kinase mutated in the ATP binding pocket (As-RSK) can use regular ATP as well as a bulkier ATP analog (A*TP-S). Use of the ATP analog induces the thiophosphorylation of the substrate. After an alkylation reaction with paranitrobenzylmesylate (PNBM), thiophosphates are converted to thiophosphate esters, which are specifically recognized by an anti-thiophosphate ester antibody. All other wild-type kinases in the cell have normal ATP binding pockets that cannot accommodate the ATP analog. (B) Identification of the gatekeeper residue in RSK2. The gatekeeper residue of RSK2 was identified by aligning the sequences of c-SRC (P00523) with those of the N-terminal kinase domain of RSKs (RSK1: Q15418, RSK2: P51812, RSK3: Q15349, RSK4: Q9UK32). Typically, the gatekeeper residue aligns with T338 of c-SRC and is preceded by two hydrophobic amino acids and followed by an acidic and another hydrophobic amino acid. RSK2 gatekeeper residue was thereby identified as L147. (C-E) HeLa cells expressing As2-RSK or WT-RSK were infected with TMEV for 8h (MOI 5) (C and E) or with EMCV for 3h30min (MOI 5) (D). Cells were then permeabilized with digitonin, and N6-Bn-ATP-γ-S was added for 1 hour. Cells were then lysed and either NUP98 (C and D) or thio-phosphate-ester containing proteins were immunoprecipitated (E). (C and D) Immunoblots showing a concentrated amount of NUP98 in the immunoprecipitation samples and thiophosphate-ester proteins. (E) Immunoblots showing NUP98 in the thiophosphate ester IP fraction when L^WT is present. Detection of viral capsid in the lysates was used as control for infection.

by RSK likely triggers a global perturbation of nucleocytoplasmic trafficking and may therefore facilitate cytoplasmic replication of cardioviruses.

## Discussion

Recent work showed that some proteins expressed by unrelated viruses and bacteria hijack host RSK kinases through a conserved DDVF linear motif that likely emerged in those proteins by convergent evolution [1,2]. Occurrence of the DDVF motif in RHBDF1, a known target of RSKs suggests that some proteins, either from pathogens or from the host cell, can associate with RSK through the DDVF motif to promote their own phosphorylation by RSKs [2]. In the case of TMEV L protein, a point mutation in the C-terminal domain of the protein (L$^{M60V}$) abrogated L activities although this mutation affected neither L-RSK interaction nor RSK activation by L [1,32]. The phenotype of this mutant supported the model of the clamp where the pathogens' protein would act to bridge RSK to specific substrates. The two models are not mutually exclusive.

Our data show that L proteins from TMEV and EMCV indeed act by redirecting part of the cellular RSK kinases toward the nuclear pore complex where RSKs directly phosphorylate FG-NUPs.

Bacteria of the genus *Yersinia* use a type III secretion system to inject Yop proteins into contacted eukaryotic cells [33]. YopM was shown to interact with RSK through a C-terminal DDVF motif [1] and with protein kinase PKNs through central leucin-rich motifs. Formation of the tripartite (YopM-RSK-PKN) complex results in the phosphorylation and activation of PKN, which requires active RSK [34,35], suggesting that RSK might also be redirected toward PKN as a specific substrate. This scenario is very likely although direct phosphorylation of PKN by RSK was not formally proven.

Another question regarding the clamp model is whether the substrates to which RSKs are redirected by the pathogens' proteins are physiological RSK substrates, the phosphorylation of which is increased by their forced association with RSKs or whether they represent unconventional RSK substrates. From the blots of analog-sensitive RSK experiments (Figs 9C–9E and S2C), it appears that basal thiophosphorylation of NUP98 or NUP214 might occur, even in non-infected cells that do not contain the L protein. However, as this work is the first work reporting the use of an analog-sensitive RSK kinase, it is difficult to formally exclude that the L147A mutation introduced in the ATP binding pocket of RSK2 did not alter the substrate specificity of the kinase. Whether or not RSK contributes to physiological phosphorylation of nucleoporins thus warrants further investigations. The current identification of substrates to which RSK kinases are retargeted is, however, unlikely to be biased by the analog-sensitive (L147A) mutation of RSK because comparison was done between cells expressing the same as-kinase, after infection with different virus mutants.

Also, given that the main binding site of pathogens' proteins is located in a surface-exposed loop of RSK, relatively remote from the catalytic site, it is unlikely that this interaction can modify the catalytic site enough to accommodate structurally divergent substrates. Thus, the role of the bridging proteins is likely to increase the frequency of encounters between RSKs and specific substrates, and possibly also to drive these kinases to specific subcellular locations.

Viruses of the *Picornaviridae* family disturb nucleocytoplasmic traffic, likely to recruit host nuclear RNA-binding proteins to the cytoplasm of the infected cells, where their genome translation and replication takes place. Perturbation of the traffic was also reported to inhibit the activation of innate immunity genes such as genes coding for interferon or chemokines, because these genes rely on the translocation of transcription factors like IRF3/7 or NFκB [16,36]. Enteroviruses encode proteases 2A$^{pro}$ and 3C$^{pro}$, which directly cleave FG-NUPs [37],

thereby opening the central channel of the nuclear pore complex [20,38]. Cardioviruses, however, induce the hyperphosphorylation of FG-NUPs, thus mimicking a process occurring during mitosis, which ends up in the dismantling of the NPCs and in the free diffusion of nuclear and cytoplasmic proteins. Here, we show that TMEV L promotes direct FG-NUPs phosphorylation by RSKs. During mitosis, FG-NUP phosphorylation was proposed to involve mostly cyclin-dependent kinase 1 (CDK1), polo-like kinase 1 (PLK1) and NIMA-related kinases [23,39]. A phosphoproteomic analysis by Kosako et al. [40] also suggested the involvement of the ERK pathway, ERK1/2 being the upstream kinase of RSK. We hypothesize that RSKs might indeed contribute, to some extent, to NUP phosphorylation during mitosis and that L triggers a stronger RSK-mediated FG-NUP phosphorylation during infection.

Our work illustrates a new virulence mechanism whereby pathogens' proteins not only activate but also redirect host kinases toward specific substrates and decipher how cardioviruses trigger RSK-mediated FG-NUP hyperphosphorylation to perturb nucleocytoplasmic trafficking in the host cell.

## Resource availability

### Lead contact

Further information and requests for resources and reagents should be directed to and will be fulfilled by the lead contact, Thomas Michiels (thomas.michiels@uclouvain.be).

### Materials availability

All reagents generated in this study are available upon request after completion of a Materials Transfer Agreement

## Material

### Cells

HeLa cells used in this work were HeLa M cells that reportedly have low endogenous RNase L activity (kindly offered by R.H. Silvermann) [41].

HeLa-RSK-DKO, (RSK1- and RSK2-deficient) and HeLa-RSK-TKO (RSK-1, RSK2- and RSK3-deficient) cells were obtained from HeLa M cells using the CRISPR-Cas9 technology [1]. As HeLa cells express very little RSK3 and virtually no RSK4 transcripts, both cell lines can be considered as RSK-KO cells. One transcribed allele is however detected carrying a 81nt in-frame deletion in RSK1 [1].

HeLa-LVX and HeLa-RSK-TKO-LVX were obtained by transduction of HeLa M cells and of HeLa-RSK-TKO cells with pLVX-EF1alpha-2xGFP:NES-IRES-2xRFP:NLS. Clones showing regular 2xGFP:NES and 2xRFP:NLS expression levels under the fluorescent microscope were selected for further use. HeLa BioID-RSK cells were obtained by transduction of HeLa-RSK-DKO cells with BLP10 and transduced cell populations were then selected with 1mg/ml of G418 (Roche). BioID2-RSK2 expression was then verified by western blot. HeLa WT-RSK, HeLa As1-RSK and HeLa As2-RSK cells were obtained by transduction of HeLa-RSK-TKO cells with TM1117, BLP20 and BLP21 respectively and transduced cell populations were selected with 1mg/ml of G418 (Roche). WT, As1 or As2-RSK expression was then tested by western blot.

HeLa-RSK-TKO cells were transduced with the empty lentiviral vector TM952 or with TM1116-19 derivatives expressing the four isoforms of human RSK. Transduced cell populations were selected with 2mg/ml of G418.

BHK-CL13 and 293T-CL13 cells were obtained by lentiviral transduction of CL13 (TMEV capsid precursor—IRES—mCherry) in the corresponding cells. mCherry-positive cells were then sorted by FACS.

293T cells were kindly provided by F. Tangy (Pasteur Institute, Paris). Both HeLa M and 293T cells and their derivatives were maintained in Dulbecco's modified Eagle medium (DMEM) (Lonza) supplemented with 10% of fetal calf serum (FBS, Sigma), 100 U/ml penicillin and 100µg/ml streptomycin (Thermo Fisher). BHK-21 cells (ATCC), used for viral production and titration, were maintained in Glasgow's modified Eagle medium (GMEM) (Gibco) supplemented with 10% newborn calf serum (Gibco), 100 U/ml penicillin, 100µg/ml streptomycin and 2.6g/l tryptose phosphate broth (Gibco). All cells were cultured at 37˚C, in 5% $CO_2$.

## Viruses

TMEV viruses used in this study are derivatives of KJ6, a variant of the persistent DA strain (DA1 molecular clone) adapted to grow on L929 cells [42]. FB09 carries the M60V mutation in L ($L^{M60V}$), which was shown to abrogate L toxicity [12,32]. MIP146 carries the F48A mutation which abrogates L interaction with RSK. KJ6, FB09 and MIP146 are indicated as TMEV $L^{WT}$, $L^{M60V}$ and $L^{F48A}$ respectively.

EMCV viruses used is this study are derived from pMC24, a Mengo virus molecular clone carrying a shortened (24C) polyC tract [43]. The virus denoted $EMCV^{WT}$ (used in Fig 9) is the virus produced from the pMC24 plasmid. Viruses denoted EMCV $L^{WT}$ and $L^{Zn}$ (used in Figs 2–4) are produced from plasmids pFS269 and pTM1098 and express N-terminally Flag-tagged L proteins. $L^{Zn}$ carries two point mutations (C19A and C22A) in the zinc finger of L, which likely affect the overall structure of the protein [44].

These viruses were produced by reverse genetics from plasmid constructions containing the full-length viral cDNA sequences. To this end, BHK-21 cells were electroporated (1500V, 25µFd, no shunt resistance) using a Gene pulser apparatus (Bio-Rad) with viral RNA produced by *in vitro* transcription (RiboMax transcription kit P1300, Promega). Supernatants were collected 48-72h after electroporation, when cytopathic effect was complete. After 2 to 3 freeze-thaw cycles, the supernatants containing the virus were clarified by centrifugation at 1258g for 20 min and viruses were stored at -80˚C. Viruses were titrated by plaque assay in BHK-21 cells.

## Trans-encapsidated viral replicons

FW12 is the replicon encoding for BioID2-HA-linker(3xGGGGS)-$L^{WT}$. This replicon derives from TMEV DA1. It was constructed by replacing the capsid-coding region of the virus by a sequence coding sequentially (5'-to-3'), in a single frame, for: BioID2-HA-linker(3xGGGGS)-L, the L/VP4 boundary cleaved by protease 3C, eGFP, the VP2 coding segment encompassing the CRE sequence, the 3' end of the 2A coding region and the following part of the viral genome (see Fig 5). The BioID-$L^{M60V}$ (FW16) and BioID-$L^{F48A}$ (FW17) viral replicons are derivates of FW12 carrying the indicated mutation. Trans-encapsidation of the replicon occurred in 293T cells transduced with the lentiviral vector CL13 to stably express the viral capsid precursor protein. pCL13, the plasmid carrying the lentiviral vector, is a pTM945 derivative expressing the 10 C-terminal aminoacids of L followed by the entire capsid precursor of virus DA1. C-terminal residues of L were inserted in the construct to restore a genuine L/VP4 cleavage site for protease 3C, which allows the capsid protein to start with an N-terminal glycine, potentially undergoing myristoylation [45]. Importantly, 4 silent mutations were introduced in the VP2 coding sequence corresponding to the *CRE* region to prevent the replication of recombinant wild type viruses that would emerge through recombination between replicon RNA and capsid-coding mRNA in packaging cells.

293T-CL13 cells were seeded (600,000/well) in 6-well plates. 24h after seeding, cells were transfected (TransIT-mRNA, Mirus) with 5µg of replicon RNA previously obtained by *in vitro* transcription. Six hours post-transfection, medium was changed to DMEM without serum. The infected cell population was passaged 8 times and reseeded with part of the collected supernatant to amplify viral production. Then, 150ml of replicon-containing supernatant was collected and filtered (0.45µm). SDS was then added to a final concentration of 0.5% and, after 2h incubation at RT, supernatants were centrifuged for 5min, at 1258g at 20˚C. Next, 13ml of clarified supernatant was added to a 2.6ml sucrose 30% cushion in polyallomer tubes (Beckman) and centrifuged for 16h at 20˚C, 25.000rpm in a SW28 swinging bucket rotor. After ultracentrifugation, pellets containing the replicons were resuspended in 150µl of 10mM Tris-HCl pH 7.5. and further dialyzed in 10mM Tris-HCl pH 7.5. Replicons were then titrated by plaque assay using BHK-21 cells stably expressing viral capsid proteins (BHK-21 CL13), and kept at -80˚C.

### Lentiviral vectors and cell transduction

- pLVX-EF1alpha-2xGFP:NES-IRES-2xRFP:NLS was a gift from Fred Gage (Addgene plasmid # 71396) [46]

Other lentiviral vectors used for protein expression were derived from pCCLsin.PPT. hPGK.GFP.pre. [47]

- TM945 is a Prom$_{CMV}$-MCS-IRES-mCherry construct [48]

- TM952 is a Prom$_{CMV}$-MCS-IRES-neo construct [49]

- BLP10 is a TM952 derivative coding for BioID2-HA-linker(3xGGGS)-MuRSK2. Note that this construct expresses murine RSK2, which differs from human RSK2 by a single amino acid.

- BLP20 is a TM952 derivative coding for As1-RSK2 (hu-RSK2 with Analog-sensitive kinase 1 (As1) mutation (Leu 147 → Gly).

- BLP21 is a TM952 derivative coding for As2-RSK2 (hu-RSK2 with Analog-sensitive kinase 2 (As2) mutation (Leu 147 → Ala).

- CL13 is a TM945 derivative expressing the 10 C-terminal aminoacids of L followed by the entire capsid precursor of virus DA1.

- TM1116, TM1117, TM1118 and TM1119 are TM952 derivatives carrying the sequences coding for 3xHA-Human RSK1, RSK2, RSK3 and RSK4 respectively [1]. They were constructed using the Gateway technology (Invitrogen) from donor plasmids Hs.RPS6KA1, Hs. RPS6KA2, Hs.RPS6KA3 and Hs.RPS6KA6 kindly provided by Dominic Esposito through the Addgene collection (Addgene refs: 70573, 70575, 70577, and 70579, respectively)

Lentiviruses were produced in 293T cells by co-transfection of the following plasmid, using TransIT-LT1 reagent (Mirus Bio): 2.5 µg of lentiviral vector, 0.75 µg of pMD2-VSV-G (VSV-glycoprotein), 1.125 µg of pMDLg/pRRE (Gag-Pol), and 0.625 µg of pRSV-Rev (Rev). DNA quantities are for transfection of 1 well of a 6-well plate. Supernatants were typically collected 72h post transfection and filtered (porosity: 0.45µM). For transduction, cells were typically seeded in a 24-well plate as of 5,000–10,000 cells/well and infected 1 or 2 times with 100µL of filtered lentivirus.

### Method details

### Biotinylation experiments

BioID-RSK: HeLa BioID-RSK cells were seeded in 6-well plates at a density of 90,000 cells per well. Two 6-well plates were used per condition. 24h after seeding the cells, the medium was changed to OptiMEM (Gibco) depleted from biotin (previously incubated with streptavidin

beads for 2h at 4˚C, then filtered). Cells were kept in OptiMEM without biotin for 48h, then they were infected with 600μl of virus per well at an MOI of 2.5. One hour post-infection, 2ml/ well of DMEM containing 5μM biotin was added. Infection proceeded for 16h. For immunostaining: 1,000 HeLa BioID-RSK cells were seeded in wells of a 96-well plate. 24h after seeding the cells, the medium was changed to OptiMEM without biotin. Cells were then incubated without biotin for 48h, then they were infected with 50μL of virus per well at an MOI of 5. One hour post-infection 150μL of DMEM containing 5μM biotin was added. Infection proceeded for 10h.

BioID-L: HeLa cells were seeded in 6-well plates as of 160,000 cells per well. Two 6-well plates were used per condition. 24h after seeding, the cells were infected with 600μl of BioID-L replicons at an MOI of 2.5. One hour post-infection 2ml/well of DMEM containing 5μM biotin was added. Infection proceeded for 14h. For immunostaining: 3,500 HeLa cells were seeded in wells of a 96-well plate. 24h after seeding, the cells were infected with 50μL of virus per well at an MOI of 5. One hour post-infection 150μL of DMEM containing 5μM biotin was added. Infection proceeded for 10h.

## Streptavidin pulldown

Cells were washed with PBS 3 times and lysed with 200μL/well of stringent lysis buffer (50mM Tris-HCl pH 7.6, 500mM NaCl, 0.4% SDS, 1mM DTT, 1 tablet of Pierce phosphatase/protease inhibitor (Thermo Scientific) per 10ml of lysis buffer) for 15 min at room temperature (RT). Lysates were then diluted twice by addition of 200μl/well of 50mM Tris-HCl pH7.6 and homogenized by 10 passages through 21G needles. Lysates were then cleared by centrifugation at 12,000g for 10 min at RT. Supernatants were then transferred to new tubes, 200μL (per condition) of protein A/G magnetic beads (Pierce) were added to remove non-specific binding and incubated for 30 min at RT. Supernatants were then transferred to a new tube and a sample of 160μL per condition was mixed with 80μL of 3x Laemmli buffer (cell lysate control). The rest of the supernatant was incubated for 2h at RT with 260μL (per condition) of Streptavidin magnetic beads (Pierce). Streptavidin beads were then washed once with 2% SDS, and twice with "normal lysis buffer" (50mM Tris-HCl pH 7.5, 100mM NaCl, 2mM EDTA, 0.5% NP40, 1 tablet of phosphatase/protease inhibitor (Thermo Scientific) per 10ml of lysis buffer) for 5 min at RT. Beads were resuspended in 40μl of 1x Laemmli buffer and heated for 5min at 100˚C to allow protein's separation from the beads. Supernatants were then conserved at -20˚C.

## Immunostaining

Cells seeded in 96-well screenstar plates (Greiner, 655866) were fixed with PBS-4% PFA for 5 min at RT. Cells were then permeabilized with PBS-0.2% Triton X-100 (ICN Biomedicals Inc.) for 5 min at RT. Cells were then blocked with TNB blocking reagent (Perkin Elmer) for 1h at RT. Antibodies were diluted in TNB blocking reagent: rabbit anti-TMEV 3D polymerase 1/ 1000 (rabbit, kindly provided by M.Brahic), anti-2A 1/1200 (rabbit, home-made), anti-NUP98 (rat, N1038, Sigma) 1/400, anti-POM121 (rabbit, 15645-1-AP, Proteintech) 1/400 or anti-PTB (mouse, 324800, Invitrogen) 1/400; and added to the cells and incubated for 1h at RT. Cells were then washed three times with PBS-0.1% Tween 20 for 5 min. Cells were then incubated with secondary antibodies (anti-rat Alexa Fluor 647, anti-rabbit Alexa Fluor 488 or anti-rabbit Alexa Fluor 594, 1/800, anti-mouse Alexa Fluor 488 1/400) or streptavidin (Alexa fluor 488 or Alexa fluor 594, 1/500) for 1h at RT. Finally, cells were washed 3 times with PBS-0.1% Tween 20 and kept in PBS-Azide 0.02% until analysis. Fluorescence analysis was made with a spinning disk confocal microscope (Zeiss) and adjustment of the images was performed using the Zen image analysis software (Zeiss). Exposure time, image brightness and contrast were adjusted

equally within an experiment. For LVX cells (qualitative analysis), brightness and contrast were adjusted using the automatic "best fit" setting of the Zen software. For Z-stack pictures, slice intervals were 0.62 μm. Colocalization analysis (intensity vs distance plots) were done with ImageJ.

## Western blotting

Proteins in Laemmli buffer were heated for 5 minutes at 100˚C. They were then run in 8 or 10% Tris-glycine SDS polyacrylamide gels and then transferred to either polyvinylidenefluoride or nitrocellulose membranes (only for Mab414 detection). Membranes were blocked with TBS-5% non-fat dry milk (Regilait) or TBS-5% BSA (for streptavidin-HRP blots) for 1 hour at RT. Primary antibodies: anti-2A (TMEV, rabbit home-made 1/4000), anti-Phospho-S380-RSK (rabbit, CST11989 –Cell signaling technology, 1/5000), anti-PKR (rabbit, 18244-1-AP–Proteintech, 1/4000), anti-Phospho-T446-PKR (rabbit, AB32036 –Abcam, 1/4000), anti-ß-actin (mouse, A5441 –Sigma, 1/10,000), anti-TMEV 3D polymerase (rabbit, kindly provided by M. Brahic, 1/2000), anti-NUP98 (rat, N1038 –Sigma, 1/2000), anti-HA (mouse, MMS101P –Covance, 1/4000), anti-RSK (BD610226 1/2000), anti-phospho-Akt-substrates: RxxS*/T* (phospho-RSK substrates, rabbit, CST9614 –Cell signaling technology, 1/1000), anti-thio-P-ester (rabbit, NBP2-67738 –Novusbio, 1/1000), anti-TMEV capsid (home-made, 1/2000), anti-FLAG M2 (mouse, Sigma, F1804 1/5000), anti-FG-NUP Mab414 (mouse, MMS120P –Covance 1/5000) were diluted in blocking solution and were incubated with membranes overnight. Membranes were then washed 3 times for 15min with TBS-0.1% Tween 20. Secondary antibodies (anti-rabbit, anti-mouse or anti-rat coupled to HRP, Dako, 1/5000) or Streptavidin-HRP (Dako, 1/1000) were then added for 1h at RT. Then, membranes were washed three times with TBS-0.1% Tween 20, once with TBS and revelation was made with SuperSignal West chemiluminescence substrate (Pico or Dura, Thermoscientific).

## Mass spectrometry

Streptavidin pulldown samples were resolved using a 10% Tris-Glycine SDS gel run until 6mm migration in the separating gel. Proteins were colored using PageBlue (Thermo Scientific, 24620). The 6mm bands containing whole proteins were cut into 3 different slices and trypsin digested. Peptides were dissolved in solvent A (0.1% TFA in 2% ACN), directly loaded onto reversed-phase pre-column (Acclaim PepMap 100, Thermo Scientific) and eluted in backflush mode. Peptide separation was performed using a reversed-phase analytical column (Acclaim PepMap RSLC, 0.075 x 250 mm,Thermo Scientific) with a linear gradient of 4%-27.5% solvent B (0.1% FA in 98% ACN) for 35 min, 27.5%-40% solvent B for 5 min, 40%-95% solvent B for 5 min and holding at 95% for the last 5 min at a constant flow rate of 300 nl/min on an EASY-nLC 1000 RSLC system. The peptides were analyzed by an Orbitrap Fusion Lumos tribrid mass spectrometer (ThermoFisher Scientific). The peptides were subjected to NSI source followed by tandem mass spectrometry (MS/MS) in Fusion Lumos coupled online to the nano-LC. Intact peptides were detected in the Orbitrap at a resolution of 120,000. Peptides were selected with the quadrupole for MS/MS using HCD setting at 35; ion fragments were detected in the ion trap. A data-dependent procedure that alternated between one MS scan followed by MS/MS scans was applied for 3 seconds for ions above a threshold ion count of 5.0E3 in the MS survey scan with 30.0s dynamic exclusion. The electrospray voltage applied was 2.1 kV. MS1 spectra were obtained with an AGC target of 4E5 ions and a maximum injection time of 50ms, and MS2 spectra were acquired with an AGC target of 1E4 ions and a maximum injection set to 35ms. For MS scans, the m/z scan range was 375 to 1800. The resulting MS/MS data was processed using Sequest HT search engine within Proteome Discoverer 2.5 SP1 against a

*Homo Sapiens* protein database obtained from Uniprot (78 787 entries) and containing the sequences of viral proteins and BioID2-RSK. Trypsin was specified as cleavage enzyme allowing up to 2 missed cleavages, 4 modifications per peptide and up to 5 charges. Mass error was set to 10 ppm for precursor ions and 0.1 Da for fragment ions. Oxidation on Met (+15.995 Da), phosphorylation on Ser, Thr and Tyr (+79.966 Da), conversion of Gln (-17.027 Da) or Glu (- 18.011 Da) to pyro-Glu at the peptide N-term, biotinylation of Lys (+ 226.077) were considered as variable modifications. False discovery rate (FDR) was assessed using Percolator and thresholds for protein, peptide and modification site were specified at 1%.

PSM Ratio calculations were as followed:

$$BioID - RSK : \frac{PSM(L^{WT})}{PSM(L^{M60V}) + PSM(mock) + 1} \qquad BioID - L : \frac{PSM(L^{WT}) + PSM(L^{F48A})}{PSM(L^{M60V}) + PSM(mock) + 1}$$

## GST-S6 recombinant production and purification

ER2566 electrocompetent bacteria were electroporated with pFS208 (plasmid coding for GST-S6) and cultured in 10mL of TSB at 37˚C O/N. Bacteria were then amplified at 37˚C in 300mL of TSB until DO600 reaches 1. At this point, we added IPTG at a final concentration of 0.2mM and let it at 37˚C for 2 hours. Then bacteria were centrifugated at 6,000rpm 15min 4˚C and resuspended in PBS, 1% Triton, 0.2mM PMSF, 2mM EDTA, 0.1% ß-mercaptoethanol. Bacteria were lysed with a French press and centrifuged at 21,000g for 20 min at 4˚C. Supernatant containing soluble protein was then kept for GST-pulldown. GST-pulldown was performed with agarose-glutathion beads. GST-S6 was eluted from the beads with 50mM Tris-HCl pH8 – 10mM glutathion and then purified by dialysis with 25mM Hepes buffer at 4˚C.

## *In vitro* kinase assay

293T cells were seeded (500,000 cells/well) in 6-well plates. 24h after seeding, cells were transfected with TransIT-LT1 reagent (Mirus Bio) with 2.5 μg of plasmids pTM1117 (coding for WT-RSK2), pBLP20 (coding for As1-RSK2) and pBLP21 (coding for As2-RSK2). 6 hours post-transfection, cells were treated with 32nM PMA over-night. HA-RSKs were then immunoprecipitated with anti-HA magnetic beads. Therefore, cells were washed once with PBS and lysed with 150μL of lysis buffer (50mM Tris-HCl pH8, 100mM NaCl, 0.5% NP40, 2mM EDTA, 1 tablet of phosphatase/protease inhibitor (Thermo Scientific) per 10ml of lysis buffer) for 15min at 4˚C. Lysates were then homogenized by 10 passages through 21G needles and cleared by centrifugation at 12,000g for 10min at 4˚C. Supernatants were then transferred to new tubes, 12,5μl (per condition) of protein A/G magnetic beads (Pierce) were added to remove non-specific binding and incubated for 30 min at 4˚C. Supernatants were then transferred to a new tube and a sample of 20μl/ condition was mixed with 10μL of 3x Laemmli buffer (cell lysate control). The rest of the supernatant was incubated with 12.5μl of HA-magnetic beads (Pierce) for 2h at 4˚C. Beads were then washed 3 times with lysis buffer for 5 min at 4˚C and resuspended in 30μl of TBS, 1mM PMSF, 1mM Na₃VO₄, 1 tablet of phosphatase/protease inhibitor (Thermo Scientific) per 10ml.

For one *in vitro* kinase reaction we put: 2.5μg of GST-S6 (recombinant substrate), 10μL of kinase buffer 5x (125mM Hepes pH 7.5, 250mM NaCl, 100mM ß-glycerophosphate, 5mM DTT, 100mM MgCl₂, 500μM Na₃VO₄), 0.5 μL of 10mM ATP (Roth, HN35.1) or A*TP analogs (BioLog), and water to a final volume of 30μL. To this we added 10μL of the immunoprecipitated RSK and incubated everything at 30˚C for 30 min with shacking. Following this, the samples were alkylated with PNBM (Abcam) at a final concentration of 2.5mM for 2 hours at room temperature with shacking. Reaction was stopped by the addition of Laemmli buffer 3x.

Samples were then heated at 100˚C for 5 min, separated from the magnetic beads and kept at -20˚C.

## Thiophosphorylation in permeabilized cells

$1.5 \times 10^6$ HeLa-RSK-TKO TM1117(WT-RSK2) or BLP21 (As2-RSK2) cells were seeded in 10-cm dishes, 1 10-cm dish per condition. The next day, cells were infected at an MOI of 5 for 8h30min (for TMEV) and 3h30min (for EMCV). Next, cells were permeabilized with 500µL of the analog-kinase buffer: 20mM Hepes pH 7.5, 100mM KOAc, 5mM NaOAc, 2mM MgOAc$_2$, 1mM EGTA, 20µg/ml digitonin, 10mM MgCl$_2$, 0.5mM DTT, 1x phosphatase inhibitor cocktail 2 (P5726, Merck), 1x cOmplete protease inhibitor (11697498001, Roche), 57µg/ml creatin kinase (Calbiochem 23895), 5mM Creatin phosphate (Calbiochem, 2380), 0.1mM ATP, 0.1 mM N6-Bn-A*TP analog (BioLog), 3mM GTP (Roth, K056.4) in order to get the ATP analog inside the cells and the thiophosphorylation reaction to happen. Reaction proceeded for 1h at 37˚C, 5% CO2 with the dishes on a rocking plate.

## Immunoprecipitation after thiophosphorylation reactions

NUP98 IP: cells were lysed by adding 500µL of 2x salty lysis buffer (100mM Tris-HCl pH8, 800mM NaCl, 2% Triton X-100, 4mM EDTA, 2mM DTT, 1 tablet of phosphatase/protease inhibitor (Thermo Scientific) per 10ml of lysis buffer) to the analog-kinase buffer for 15 min at 4˚C. Lysates were then diluted 4x by addition of 3ml of regular lysis buffer (50mM Tris-HCl pH8, 100mM NaCl, 0.5% NP40, 2mM EDTA, 1 tablet of phosphatase/protease inhibitor (Thermo Scientific) per 10ml of lysis buffer). Lysates were then homogenized by 10 passages through 21G needles and cleared by centrifugation at 12,000g for 10min at 4˚C. Supernatants were then transferred to new tubes and 30µL (per condition) of protein A/G magnetic beads (Pierce) were added to remove non-specific binding and incubated for 30 min at 4˚C. Supernatants were then transferred to a new tube and a sample of 200µL/ condition was alkylated with PNBM at a final concentration of 2.5mM for 2h at RT. Then alkylation reaction was stopped by addition of 3x Laemmli buffer (cell lysate control). The rest of the supernatant was incubated O/N at 4˚C with 8µg (per condition) of anti- NUP98 (rat, N1038 –Sigma). 62.5µl of A/G beads per condition were added and incubated for 2h at 4˚C. A/G beads were then washed three times with regular lysis buffer for 5 min at 4˚C. Beads were resuspended in 40µl of kinase buffer 1x (see above: in vitro kinase assay). The IP samples were then alkylated with PNBM at a final concentration of 2.5mM for 2h at RT. Alkylation reaction was stopped by the addition of 3x Laemmli, and immunoprecipitated proteins were separated from the beads after heating 5min at 100˚C.

Thiophosphate ester IP: cells were lysed by adding 500µL of 2x lysis buffer (100mM Tris-HCl pH8, 200mM NaCl, 1% NP40, 40mM EDTA, 1 tablet of phosphatase/protease inhibitor (Thermo Scientific) per 10ml of lysis buffer) for 15min at 4˚C. Lysates were then homogenized by 10 passages through 21G needles and cleared by centrifugation at 12,000g for 10min at 4˚C. Supernatants were then alkylated with PNBM at a final concentration of 2.5mM for 1h30min at RT. As PNBM inhibits immunoprecipitation, lysates were run through PD10 columns to remove PNBM prior to the immunoprecipitation procedure. Fractions containing the proteins were kept and incubated with 30µl per condition of A/G magnetic beads (to avoid non-specific binding) for 30 min at 4˚C. Supernatants were then transferred to a new tube and 1/10th of the lysate (200µl) was mixed with 100µl of 3x Laemmli (cell lysate control). The rest of the supernatant was incubated with 8µg per condition of anti-thiophosphate ester antibody (rabbit, NBP2-67738 –Novusbio) for 2h at 4˚C. Then 62.5µl per condition of A/G magnetic beads were added and incubated for 2h at 4˚C. A/G beads were then washed three times with regular

lysis buffer for 5 min at 4˚C and resuspended in 60μl of Laemmli 1x. Immunoprecipitated proteins were separated from the beads after heating 5min at 100˚C.

## Quantification and statistical analysis

### Data processing and statistical analysis (for MS)

Protein identification and label-free quantitation were performed with MaxQuant version 1.6.7.0 [50]. Database searching was performed against the UniProt FASTA database, using a false-discovery rate at the peptide and protein level was set to 0.01 and allowing a maximum of two missed cleavages. All search parameters are available in the parameter file available in the study repository (https://github.com/UCLouvain-CBIO/2022-RSK-Nups-VIRO). After filtering out of identified contaminants, identified reverse sequences and proteins having missing values in all samples, Max Quant protein intensities were log-2 transformed and normalized using sample median alignment. In order to cope with the numerous drop-outs in protein intensities, the proDA method, as described in [28] was applied.

For the analysis of the BioID-L experiment, the backbone linear model included one indicator variable representing a potential batch effect between 2 groups of replicates, and 2 condition indicator variables, representing the effect of resp. WT and F48A conditions with respect to the reference condition, i.e. M60V condition. Potential batch effect was tested for each protein using two-sided Wald test on the batch indicator variable. Since adjusted p-values were $> 0.2$ for all proteins, this indicator variable was subsequently removed from the model. Then, the effects of WT and F48 conditions (w.r.t. M60V) were tested, for each protein, using one-sided Wald tests, since the anticipated sign of the effect was known. All p-values were adjusted using Benjamini-Hochberg corrections [51].

For the analysis of the BioID-RSK experiment, the backbone linear model included one indicator variable representing a potential batch effect between 2 groups of replicates, and 1 condition indicator variable, representing the effect of WT condition with respect to the reference condition, i.e. M60V condition. Potential batch effect was tested for each protein using two-sided Wald test on the batch indicator variable. Since adjusted p-values were $> 0.6$ for all proteins, this indicator variable was subsequently removed from the model. Then, the effect of WT condition (w.r.t. M60V) was tested, for each protein, using a one-sided Wald test, since the anticipated sign of the effect was known. All p-values were adjusted using Benjaminin-Hochberg corrections.

### Statistical analysis on IF

Statistical analysis on immunofluorescence experiments was done using GraphPad Prism v9. One-way ANOVA with Tukey's multiple comparisons tests were used to compare all samples with each other. The number of independent experiments (n) and statistical comparison groups are indicated in the Figures and Figure legends. (* $p< 0.05$, ** $p<0.01$, *** $p<0.001$, **** $p<0.0001$).

## Supporting information

**S1 Fig. BioID-RSK and BioID-L fusion proteins keep their normal activities.** (A) L protein activities are conserved in HeLa BioID-RSK cells. HeLa BioID-RSK cells were infected with $L^{WT}$ and $L^{M60V}$ viruses for 16h. Western blot showing BioID-RSK activation by the $L^{WT}$ and $L^{M60V}$ proteins (p-RSK at S380) and PKR inhibition by $L^{WT}$ but not by $L^{M60V}$ (p-PKR at T446 is a marker of PKR activation). (B) BioID-L fusion proteins maintain their corresponding activities. Immunoblots show the detection of activated RSK (p-S380), activated PKR (p-T446)

and Nup98 hyperphosphorylation (migration shift) in HeLa cells infected for 16h with BioID-L$^{WT}$, BioID-L$^{M60V}$, BioID-L$^{F48A}$ replicons (2 lanes each) and, as a control, with L$^{WT}$ and L$^{M60V}$ viruses (1 lane each). Viral polymerase 3D and ß-actin were detected as infection and loading controls respectively.
(TIF)

**S2 Fig. Analog-sensitive RSKs (As1-RSK and As2-RSK) keep their kinase activity and thio-phosphorylate substrates in vitro.** (A) Analog-sensitive RSKs (As1 and As2-RSKs) recue L protein activities. HeLa RSK-TKO cells transduced with an empty vector (-) or with lentiviral vectors expressing WT-RSK, As1-RSK or As2-RSK, were infected (MOI 2.5) for 15h with L$^{WT}$ or L$^{M60V}$ viruses. Western blots show the detection of HA-(RSK), P-PKR (T446), PKR, NUP98, 3D viral polymerase as a control of infection and ß-actin as a loading control. RSKs activation (p-RSK S380) PKR inhibition (inhibition of pPKR T446) and NUP98 hyperphosphorylation (shift upwards) in As-RSK expressing cells paralleled those observed in WT-RSK expressing cells. (B) GST-S6 (thio)-phosphorylation by As1, As2 and WT RSKs in an in vitro kinase assay. 293T cells were transfected with plasmids coding for WT-RSK, as1-RSK or as2-RSK. 6 hours post-transfection, cells were treated with phorbol myristate acetate (PMA) to activate RSKs. 18 hours later, RSKs were immunoprecipitated with an anti-HA antibody. An in vitro kinase assay was performed with the immunoprecipitated RSKs, GST-S6 (recombinant substrate) and either ATP or N6-Bn-ATP-γ-S or N6-PhEt-ATP-γ-S. Reaction proceeded for 30min at 30°C before alkylation by PNBM for 2h at room temperature. Reaction was stopped by addition of sample buffer. Samples were analyzed by western blot with antibodies against HA-(RSK), RxxS*/T* (antibody against phosphorylated RSK substrates; here: phospho-GST-S6), and against thiophosphate ester. WT RSK was able to use ATP but was not able to use the ATP analogs to phosphorylate GST-S6 (RxxS*/T*). As1 and As2-RSK were able to use ATP to phosphorylate GST-S6 (RxxS*/T*) but were also able to use both ATP analogs, with a preference for N6-Bn-ATP-γ-S (Thiophosphate ester). Dashed lines between lanes indicate deletion of irrelevant lanes from the same membrane. (C) Thiophosphorylation of NUP214 by As2-RSK. Immunoblots showing NUP214 in the thiophosphate ester IP fraction when L$^{WT}$ is present. HeLa cells expressing As2-RSK or WT-RSK were infected with TMEV for 8h (MOI 5). Cells were then permeabilized with digitonin, and N6-Bn-ATP-γ-S was added for 1 hour. Cells were then lysed and thiophosphate-ester containing proteins were immunoprecipitated.
(TIF)

## Acknowledgments

We are grateful to Melissa Drappier for suggesting the analog-sensitive kinase experiments. We are much indebted to Kevan Shokat for quick support and detailed protocols concerning this great technique. We thank Frank J.M. van Kuppeveld for the gift of anti-EMCV capsid antibodies. We thank Stéphane Messe for excellent technical assistance, Nicolas Dauguet for expert help in cell sorting, Patrick Van Der Smissen for help in confocal microscopy, and Eric Freundt and Frank J.M. van Kuppeveld for critical reading of this manuscript.

## Author Contributions

**Conceptualization:** Belén Lizcano-Perret, Thomas Michiels.

**Formal analysis:** Philippe Hauchamps, Frédéric Sorgeloos, Laurent Gatto.

**Funding acquisition:** Thomas Michiels.

**Investigation:** Belén Lizcano-Perret, Cécile Lardinois, Fanny Wavreil, Gaëtan Herinckx, Didier Vertommen, Thomas Michiels.

**Methodology:** Belén Lizcano-Perret, Cécile Lardinois, Frédéric Sorgeloos, Didier Vertommen, Thomas Michiels.

**Project administration:** Thomas Michiels.

**Supervision:** Thomas Michiels.

**Writing – original draft:** Belén Lizcano-Perret.

**Writing – review & editing:** Belén Lizcano-Perret, Frédéric Sorgeloos, Didier Vertommen, Thomas Michiels.

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
