## [Decision Letter · Decision Letter 0]

26 Oct 2022

Dear Prof Michiels,

Thank you very much for submitting your manuscript "Cardiovirus leader proteins retarget RSK kinases toward alternative substrates to perturb nucleocytoplasmic traffic" for consideration at PLOS Pathogens. As with all papers reviewed by the journal, your manuscript was reviewed by members of the editorial board and by several independent reviewers. The reviewers appreciated the attention to an important topic.

Based on the reviews, we are likely to accept this manuscript for publication, providing that you modify the manuscript according to the review recommendations, with a particular attention to important questions raised by reviewer 1.

Please also improve the quantitation of microscopy data which was requested by more than one reviewer

Sincerely,

George A. Belov, PhD

Associate Editor

PLOS Pathogens

Alexander Gorbalenya

Section Editor

PLOS Pathogens

Kasturi Haldar

Editor-in-Chief

PLOS Pathogens

orcid.org/0000-0001-5065-158X

Michael Malim

Editor-in-Chief

PLOS Pathogens

orcid.org/0000-0002-7699-2064

Reviewer Comments (if any, and for reference):

Reviewer's Responses to Questions

**Part I - Summary**

Reviewer #1: The manuscript by Lizcano-Perret et al describes a novel mechanism by which viruses promote their replication and reduce antiviral immunity, namely via proteins that act as adaptor proteins to retarget host cell kinases towards specific substrates. As an example of this ‘model of the clamp’, they show that cardioviruses of the family picornaviridae code for non-structural proteins (Leader proteins) that can clamp both the host protein kinase RSK and a host substrate, nucleoporins. This enforces hyperphosphorylation of the nucleoporins, thereby triggering nucleocytoplasmic trafficking perturbation, which supports viral replication. The study encompasses several high-end methodologies to dissect the role of host and virus proteins in this process, such as a reciprocal BioID proxeosome determination to identify substrates binding to the Leader-RSK complex and an analog-sensitive kinase system. The data are convincing and the manuscript is well written. I have a few requests for further clarification, which I specify below.

Reviewer #2: This study examines the mechanisms governing ribosomal S6 kinase (RSK) activity and substrate selection in cardiovirus-infected cells. TMEV expressing wild type L protein disrupts nucleo-cytoplasmic trafficking in HeLa cells as evidenced by relocalization of PTB, GFP-NES and RFP-NLS and phosphorylation of Nup98. Viruses expressing a mutant L protein (M60V) lose this ability. KO of RSK prevents relocalization and Nup98 phosphorylation and this can be rescued by expression of any RSK isoform. Similar results are obtained with EMCV expressing WT L or L with a mutant zinc finger domain. Fusion of a BioID tag to RSK, WT L or M60V L is used to identify proteins that interact with L protein and are substrates for the RSK. Immunoprecipitation and mass spec analysis of biotinylated proteins reveals significant binding to FG-Nups, particularly Nup98 and this is reduced in cells expressing M60V. Confocal microscopy reveals that biotin labeled proteins are at the nuclear rim, but only if Lwt is also expressed. To identify RSK substrates, a mutant form of RSK is generated that utilizes N6-alkylated ATP-gamma-S for phosphorylation. Immunopercipitation of Nup98 revealed the presence of thiophosphorylated forms in cells infected with TMEV expressing WT L but not M60V, suggesting that Nup98 is an L-dependent substrate for RSK in infected cells.

This is a well done study that significantly adds to our understanding of how L modulates the kinase activity of RSK in infected cells. Convincing data is presented that L serves as a scaffold for binding to RSK and substrates, such as FG-Nups, so that they can be phosphorylated. The work suggests a model where RSK bound to L is targeted to the nuclear pore complex via binding of L to FG-Nups, where it phosphorylates these proteins and disrupts trafficking between the nucleus and cytoplasm. Given that other viruses and even bacteria also usurp RSK activity in cells, these findings are broadly relevant to those studying host-pathogen interactions.

Reviewer #3: The Cardiovirus leader proteins retarget RSK kinases toward alternative substrates to perturb nucleocytoplasmic traffic manuscript by Belé Lizcano-Perret, et. al conveys that p90-RSKs can be recruited to a given substrate by Cardiovirus leader proteins. The authors use Bio-ID to determine RSK binding partners as a novel strategy and show that RSK phosphorylation of nuclear pore proteins disrupts nuclear trafficking by microscopy. However, additional controls are required to adequately interpret data in the manuscript.

**Part II – Major Issues: Key Experiments Required for Acceptance**

Reviewer #1: Figure 6 and 7:

A more quantitative image analysis approach should be conducted to prove that the proteins biotinylated by BioID-RSK colocalize with the nuclear membrane/nucleoporins (eg by showing profile intensity plots (intensity vs distance plots) overlaid for both biotinylated proteins and nucleoporin fluorescence signals). This should also clarify that it seems that there is similar co-localization for WT-L and M60V-L, e.g. that there are quantitative differences. The authors should also explain why for Fig 6 the costaining is with NUP98, while in Fig 7 it is with POM12.

Fig 6:

As additional proof for Leader-driven localization of RSK to the nuclear membrane, the authors should show the co-localization of biotinylated proteins for the condition with F48A-L (RSK substrate biotinylation without Leader binding).

Reviewer #2: (No Response)

Reviewer #3: • Microscopy images demonstrating changes in nuclear-cytoplasmic diffusion (Fig 2A, Fig 2C, Fig 2E, Fig 2G, and Fig 3A, Fig 3C, Fig 3E) are difficult to interpret. Confocal microscopy eliminates out-of-plane signal, so results may not adequately capture all diffused protein, thereby skewing results. This is particularly problematic for images shown in Fig 3, which are shown without an accompanying nuclear stain. 3D reconstruction or markers are required to control for variation in imaging height in confocal microscopy. Furthermore, accompanying quantitation of protein diffusion is inadequate. To provide a percent count of diffusion-positive cells, at least 100 infected cells per field should be evaluated and quantitative cutoffs for positive and negative cells should be provided. As an alternative, authors could quantitate fluorescence to present the nuclear to cytoplasmic ratio of protein localization.

• Experiments utilizing RSK knockout and overexpression require additional controls for interpretation. In Figs 2C-D, Figs 2G-H, and Figs 3A-C, authors knockout or overexpress RSK isoforms. Expression levels of RSK isoforms should be shown to demonstrate efficacy of knockout and overexpression. Furthermore, the effect of RSK knockout or overexpression on TMEV replication is not apparent, as differences in 3D expression shown in Fig 2D are marginal. Results should be quantitated.

• Results demonstrating localization of Bio-ID-RSK targets at the nuclear periphery shown in Figs 6 and 7 are not convincing. Nuclear fluorescence from biotinylated proteins is too high to evaluate small changes on the nuclear periphery. Selective clearing to reduce fluorescence within the nucleus would aid in evaluation of the nuclear periphery. As above, quantitative criteria for positive and negative cells should be provided and more cells should be evaluated per sample.

• Additional controls are required to evaluate results shown in Fig 8. The Shokat method has not been reported for RSK2, and it is unknown whether mutation of the RSK2 ATP binding pocket changes the RSK2 substrate repertoire. Additional controls are required to demonstrate that RSK2 substrate binding is not altered .

• Statistical methods used require clarification. One-way ANOVA cannot be used to evaluate statistical differences between individual groups, yet p-values between experimental groups are reported in multiple graphs. Post-hoc tests are required to make individual comparisons when more than two groups are evaluated.

**Part III – Minor Issues: Editorial and Data Presentation Modifications**

Reviewer #1: Table in fig 5B:

please provide insight in why the BioID-RSK experiments did not yield any FG-NUP hit that reached statistical significance in the fold change L-WT over L-M60V, whereas several of the NUPs reach significant p-values in the BioID-L experiments. What does this say about the methodology, or about how the RSK-Leader complex could bind substrates?

In connection to the above question, can the authors provide potential explanations for why NUP62 and NUP214 show a much higher abundance ratios in BioID-L than in BioID-RSK?

Reviewer #2: One question that I think merits comment is why the BioID-RSK experiments didn’t show as significant an association with the FG-Nups as the BioID-L and M60V experiments did.

Reviewer #3: (No Response)

PLOS authors have the option to publish the peer review history of their article (what does this mean?). If published, this will include your full peer review and any attached files.

Reviewer #1: No

Reviewer #2: **Yes: **Kurt E Gustin

Reviewer #3: No

Figure Files:

Data Requirements:

Reproducibility:

References:

---

## [Editor Report · Decision Letter 1]

1 Dec 2022

Dear Prof Michiels,

We are pleased to inform you that your manuscript 'Cardiovirus leader proteins retarget RSK kinases toward alternative substrates to perturb nucleocytoplasmic traffic' has been provisionally accepted for publication in PLOS Pathogens.

Best regards,

George A. Belov, PhD

Academic Editor

PLOS Pathogens

Alexander Gorbalenya

Section Editor

PLOS Pathogens

Kasturi Haldar

Editor-in-Chief

PLOS Pathogens

orcid.org/0000-0001-5065-158X

Michael Malim

Editor-in-Chief

PLOS Pathogens

orcid.org/0000-0002-7699-2064
---

## [Editor Report · Acceptance letter]

8 Dec 2022

Dear Prof Michiels,

We are delighted to inform you that your manuscript, "Cardiovirus leader proteins retarget RSK kinases toward alternative substrates to perturb nucleocytoplasmic traffic," has been formally accepted for publication in PLOS Pathogens.

Best regards,

Kasturi Haldar

Editor-in-Chief

PLOS Pathogens

orcid.org/0000-0001-5065-158X

Michael Malim

Editor-in-Chief

PLOS Pathogens

orcid.org/0000-0002-7699-2064